# The impact of surface melt rate and catchment characteristics on Greenland Ice Sheet moulin inputs

Tim Hill[1, *] and Christine F. Dow[1, 2]

[1]Department of Applied Mathematics, University of Waterloo, Waterloo, Canada
[2]Department of Geography and Environmental Management, University of Waterloo, Waterloo, Canada
[*]Current address: Department of Earth Sciences, Simon Fraser University, Burnaby, Canada

**Correspondence:** Tim Hill (tim_hill_2@sfu.ca)

**Abstract.** The supraglacial drainage system of the Greenland Ice Sheet, in combination with surface melt rate, controls the rate of water flow into moulins, a major driver of subglacial water pressure. We apply the Subaerial Drainage System (SaDS) model, a physically-based surface meltwater flow model, to a $\sim 20 \times 27 \text{ km}^2$ catchment on the southwestern Greenland Ice Sheet for four years of melt forcing (2011, 2012, 2015, and 2016) to (1) examine the relationship between surface melt rate and the rate, diurnal amplitude, and timing of surface inputs to moulins, (2) compare SaDS to contemporary models, and (3) present a framework for selecting appropriate supraglacial drainage models for different modelling objectives. We find that variations in the rate and timing of modelled moulin inputs related to the development of supraglacial channels are relatively more important in years with low melt volumes than years with high melt volumes. We suggest that a process-resolving supraglacial hydrology model (e.g., SaDS) should be considered when modelling outcomes are sensitive to sub-diurnal and long-term seasonal changes in the rate of discharge into moulins.

## 1 Introduction

The supraglacial drainage system acts as a mediator in the relationship between surface melt and subglacial water pressure (e.g., Bartholomew et al., 2012; Sole et al., 2013; Andrews et al., 2014; Smith et al., 2021) since meltwater generated at the surface must first flow through supraglacial catchments before being transported through the depth of the ice via moulins and crevasses. Since supraglacial drainage in part controls the form of inputs to moulins, it has the potential to significantly affect the relationship between surface melt and ice velocity (e.g. Banwell et al., 2016; Yang et al., 2020).

Water flow through the supraglacial drainage system is understood to occur through both distributed, or hillslope/interfluve, drainage (e.g. Pitcher and Smith, 2019), and through discrete supraglacial channels. The density and size of supraglacial channels is a key control on the amplitude and timing of moulin inputs, since transport through channels is more efficient (i.e., faster) than through distributed drainage (Yang et al., 2018; Pitcher and Smith, 2019; Yang et al., 2022). The evolution of the supraglacial channel network is controlled by processes including downward incision of supraglacial streams by frictional potential energy dissipation and shortwave radiation penetration (Pitcher and Smith, 2019), aspect-dependent erosion of exposed channel walls (St Germain and Moorman, 2019), and ablation of the ice surface according to the local energy balance (St Germain and Moorman, 2019). Supraglacial drainage is made more complex by supraglacial lakes, which form in topographic

depressions and may drain by slow overtopping once the lake level reaches the minimum outlet elevation (e.g., Chudley et al., 2019), rapid incision of outlet channels (e.g., Kingslake et al., 2015), or rapidly through hydrofracture at the lake bed (e.g., Das et al., 2008).

Water flow through the supraglacial drainage system characteristically acts to reduce the diurnal amplitude and delay the timing of moulin inputs relative to the diurnal cycle of surface melt (e.g., Smith et al., 2017; Muthyala et al., 2022). This behaviour
has previously been captured by supraglacial water flow models based on explicit flow routing and cascading supraglacial lake filling (Banwell et al., 2012; Koziol and Arnold, 2018), the transient filling of supraglacial lakes (Leeson et al., 2012), and flow through predefined channel networks (Yang et al., 2018; Gleason et al., 2021; Yang et al., 2022).

Each of these models makes a tradeoff between process representation and computational cost. Here, we apply the physics-based Subaerial Drainage System (SaDS) model, which aims to include a wide variety of processes (at the expense of compu-
tational efficiency), including the seasonal evolution of the channel network (Hill and Dow, 2021). The process-based formulation results in dynamic moulin inputs that are driven by changes in surface melt rate and previous melt conditions through the configuration of incised supraglacial channels.

We use SaDS to investigate the behaviour of modelled supraglacial drainage in southwest Greenland in order to: (1) explore the impacts of seasonally and annually varying melt volumes on moulin input rates, (2) compare SaDS to a suite of contempo-
rary models, and (3) provide suggestions about how to select appropriate supraglacial drainage models for different modelling objectives.

## 2   Data and Methods

### 2.1   Study Site

We model meltwater flow across a $\sim 20 \times 27$ km$^2$ portion of the southwestern Greenland Ice Sheet, centered on $67.639°$N,
$48.960°$W (Fig. 1). The site contains seven internally drained supraglacial catchments varying in surface area from <1 km$^2$ to >100 km$^2$ (Table 1). The surface elevation of the modelled domain ranges from approximately 1150 m to 1450 m, and the area falls entirely within the bare-ice ablation zone. This area has been the site of previous supraglacial hydrology mapping (Yang and Smith, 2016) and modelling (Hill and Dow, 2021), which have shown that an extensive system of meltwater channels and lakes forms each summer. The four largest lakes in the domain are labelled L1 to L4, and range in elevation from 1215 m to
1387 m and in maximum modelled surface area from 0.45 km$^2$ to 1.39 km$^2$ (Table 1).These lakes are small to average size for this portion of western Greenland, e.g. Johansson et al. (2013) report median lake sizes between 1.41 to 2.12 km$^2$ for west Greenland.

### 2.2   Model

We model water flow across the domain and meltwater inputs to moulins using the Subaerial Drainage System (SaDS) model
(Hill and Dow, 2021). SaDS is a physics-based model that combines distributed hillslope flow with channelized flow in discrete

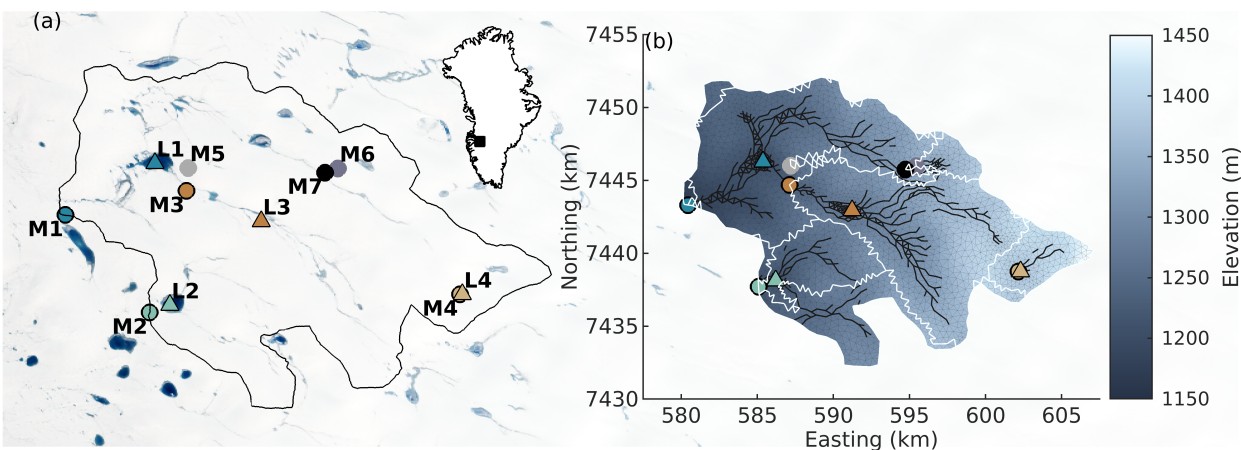

**Figure 1.** (a) Study domain with moulin locations (circles) previously mapped from a Landsat 8 image (Yang and Smith, 2016) and the four largest supraglacial lakes (triangles; L1–L4). Moulins indicated with a black outline (M1–M4) drain one of the four labelled supraglacial lakes. Background image is a Landsat 8 scene from 8 July 2021. Inset shows location of study site within the Greenland Ice Sheet. (b) Model domain with surface elevation derived from 32 m-resolution ArcticDEM mosaic data (Porter et al., 2018), modelled supraglacial channel network (black line segments), computational mesh (thin black triangles), and supraglacial sub-catchments for each moulin (white lines). Moulins (circles) and lakes (triangles) are as in (a). Coordinates are in UTM 22N projection.

**Table 1.** Catchment characteristics and catchment-averaged melt rates for the seven sub-catchments identified in Fig. 1. For catchments 1–4 that contain labelled supraglacial lakes, lake area and lake elevation are computed from modelled water depth fields. Catchments 5–7 do not contain labelled supraglacial lakes.

| Catchment | Lake Area (km$^2$) | Lake Elevation (m asl) | Catchment Area (km$^2$) | Mean elevation (m asl) | Mean melt (m w.e.) 2011 | 2012 | 2015 | 2016 |
|---|---|---|---|---|---|---|---|---|
| 1 | 1.39 | 1215 | 102.1 | 1258 | 2.39 | 2.71 | 1.28 | 2.44 |
| 2 | 0.45 | 1218 | 27.8 | 1264 | 2.40 | 2.73 | 1.24 | 2.49 |
| 3 | 1.04 | 1276 | 94.4 | 1335 | 2.05 | 2.41 | 1.00 | 2.06 |
| 4 | 0.49 | 1387 | 18.8 | 1408 | 1.66 | 2.02 | 0.82 | 1.57 |
| 5 | N/A | N/A | 1.70 | 1278 | 2.34 | 2.67 | 1.22 | 2.41 |
| 6 | N/A | N/A | 7.73 | 1366 | 1.95 | 2.35 | 0.92 | 1.96 |
| 7 | N/A | N/A | 0.73 | 1345 | 2.08 | 2.46 | 0.99 | 2.11 |

supraglacial streams. Since the flow velocity is dictated by the hydraulic potential of the water surface, lakes naturally form in topographic depressions (e.g., Leeson et al., 2012). The model explicitly computes changes to supraglacial stream cross-section area and channel network capacity based on the balance between stream incision by frictional melt along the wetted perimeter and ablation of the adjacent ice surface in order to capture seasonal changes in moulin inputs. The density of supraglacial

channels therefore changes as individual channel elements melt out if stream incision is insufficient to balance surface ablation. Flow is exchanged between the distributed and channelized drainage systems through an explicit mass exchange term that is proportional to the hydraulic potential difference between the two systems. The model is posed on an unstructured triangular mesh, with hillslope flow across elements, channelized flow on edges, and moulins on nodes. The primary model outputs are the rate of supraglacial discharge into moulins, the water depth in supraglacial channels and in the distributed system, and the incised cross-sectional area of supraglacial channels. For more details about the model, refer to Appendix A and Hill and Dow (2021). To apply SaDS to our domain, we use a mesh with 1984 nodes, 3768 elements, 5751 edges, a maximum element area of $0.15 \text{ km}^2$, and a minimum area of $0.015 \text{ km}^2$ (Fig. 1).

## 2.3 Data

Surface elevations of each element in our triangular numerical mesh are computed from ArcticDEM 32 m-resolution mosaic data (Porter et al., 2018). We first smooth the ArcticDEM with a moving average filter with an edge length of 1.44 km, and then average the pixels that lie within each triangular element to define the centroid elevation. This smoothing is required to achieve numerical convergence within the SaDS model. It is possible that moulin inputs would change with higher resolution surface elevation data. However, it does not appear that the topography has not been overly smoothed here, as evidenced by the persistence of supraglacial lakes and topographically controlled drainage pathways.

Moulin locations are adjusted from the Landsat-derived map of Yang and Smith (2016) to correct for potential discrepancy in in-situ flow routing between the acquisition time of the Landsat scene used to map hydrology features (19 August 2013; Yang and Smith, 2016) and the acquisition time of ArcticDEM (2011–2017; Porter et al., 2018). Some evidence of this type of discrepancy is provided by our sub-catchment boundaries in Fig. 1b that do not all extend to the domain boundary, which instead follows the catchment boundaries from Yang and Smith (2016).

Surface melt is computed for our domain from RACMO2.3p2 surface melt data at 5.5 km horizontal resolution and 3 hour temporal resolution (Noël et al., 2019). Melt rates from RACMO tiles are interpolated in time and linearly interpolated in space to the triangular mesh and to run the model with a timestep of 20 s. Based on tests applying RACMO melt with linear and piecewise cubic modified Akima interpolation (Akima, 1970) (implemented with the MATLAB option 'makima'), the discrepancy in the timing of peak moulin inputs is <0.5 hours and the discrepancy in the magnitude of peak moulin inputs is ~1% between linear and Akima interpolation (Fig. B1). Therefore, since the model outputs are only weakly sensitive to the interpolation scheme, and since we do not have information on how melt rate varies on timescales shorter than the RACMO timestep (3 hours), we use the simplest numerically feasible option and linearly interpolate melt rates in time. We model the years 2011, 2012, 2015, and 2016 to explore how the duration, intensity, and variability of surface melt impact the supraglacial drainage system.

## 2.4 Model initialization

In all years, SaDS is initialized with zero water depth across the supraglacial drainage system and with a bare-ice surface. This initialization neglects the impact of early season snow cover and interannual water storage in supraglacial lakes (Law et al.,

2020) on moulin inputs. While seasonal snow cover persists, surface meltwater should be retained within the snowpack, with lateral transport occurring within saturated layers within the snowpack (e.g., Colbeck, 1974), therefore delaying early season moulin inputs. A physics-based snowpack and firn model (e.g. Meyer and Hewitt, 2017) is an attractive candidate to combine with SaDS. Such a model would allow SaDS to be applied across a larger elevation range and more robustly in the early melt season. However, since this type of snow model would further increase the computational cost and significantly complicate the input data requirements of SaDS, this coupling is not pursued here.

The depth and location of supraglacial channels is spun up using the melt year in our series with average total surface melt (2013) to naturally form an initial channel network. This channel configuration is used as the initial condition for channel depth for 2011 and 2015. The channel depth from the end of 2011 and 2015 are used as initial conditions for 2012 and 2016, respectively. We expect that incorporating memory of past melt seasons in this way is key to capturing important interannual changes in moulin inputs (Hill and Dow, 2021).

## 3   Results

From model outputs for 2011, 2012, 2015, and 2016, we extract the rate of surface inputs to moulins (Fig. 2), the relative diurnal amplitude of inputs to moulins (measured as the peak-to-peak range in moulin inputs normalized by the melt season-averaged moulin input rate; Fig. 3), the lag time between local solar noon (15:22) and peak moulin inputs (Fig. 4), and the water depth in the four largest supraglacial lakes (Fig. 5).

For all seven catchments, moulin inputs generally track surface melt rate (Fig. 2), but with diminished diurnal amplitude relative to the amplitude of the surface melt rate (Fig. 3) and a phase lag of $\sim$2 to $\sim$8 hours (Fig. 4e, f). In all four years, the rate of surface inputs to moulins is closely tied to surface melt rate (Fig. 2), with a range of 0 to $\sim$70 m$^3$ s$^{-1}$ depending on catchment size and surface melt rate (Table 1).

The relative diurnal amplitude of moulin inputs varies between $\sim$15% and >90%. The diurnal amplitude shows some sensitivity to surface melt rate (Fig. 3). For example, the relative diurnal amplitude for the highest melt year (e.g., 10–30% for M1–M4 in mid-July 2012 with surface melt rate >0.07 m w.e. day$^{-1}$) is typically less than for the lowest melt year (e.g., >30–60% for M1–M4 in July 2015 with surface melt <0.06 m w.e. day$^{-1}$).

Peak moulin input rates lag behind solar noon by $\sim$2.5 to 7.5 hours. The time lag does not show clear trends with volume of surface melt rate, since a similar magnitude and range of lag time is observed in all years (Fig. 4). However, the lag time appears sensitive to rate of change of surface melt rate, with the highest lag times observed during transient peak melt events, especially with melt greater than 0.04 m w.e. day$^{-1}$ (e.g., July 2011, July 2012, June 2016).

The four labelled supraglacial lakes fill quickly in the early melt season. Once the lakes have filled (the storage volume in the absence of surface melt can be approximated by the lake depth at the end of the melt season), lake depth remains dynamically controlled by surface melt rate and by the hydraulic potential in the channel draining the lake (Fig. 5). Our model outputs show that only two lakes retain a significant depth of water at the end of the melt season (L1 and L2), while L3 and L4 drain nearly completely once melt ceases.

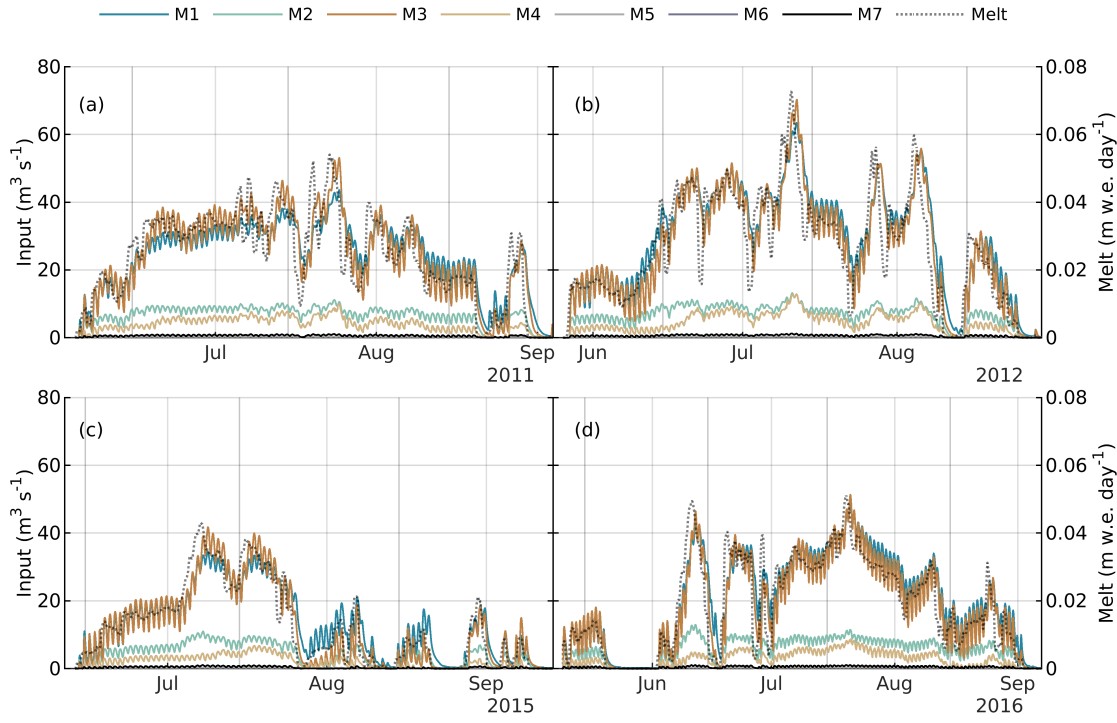

**Figure 2.** Surface inputs to moulins M1–M7 and domain-averaged 24-hour rolling mean RACMO2.3p2 surface melt rate (right axis, dotted grey curve) for (a) 2011, (b) 2012, (c) 2015, and (d) 2016.

The three small, lake-free catchments (M5, M6, M7; Area $\leq 8$ km$^2$) tend to have a larger diurnal amplitude as a fraction of mean moulin inputs and a shorter lag time than the large catchments with lakes (M1, M2, M3, M4; Area $\geq 19$ km$^2$). From June to August 2012, for example, the diurnal amplitude for catchments M5–M7 is between 50–80% and only 20–50% for catchments M1–M4. For the same period, the lag time for M5–M7 is consistently 1–3 hours less than for M1–M4 (Fig. 3b, 4b). The separation is less clear for August–September 2015, where recurring periods with no surface melt seem to obscure the trend (Fig. 3c, 4c).

Figures 2–5 suggest that surface melt is an important explanatory variable for the rate and diurnal amplitude of surface inputs to moulins, lag time between solar noon and peak moulin input, and lake water level. The extent to which surface melt rate controls these features can be quantified by comparing the coefficient of determination, $R^2$, between melt rate and each of the moulin input rate, diurnal amplitude, lag time, and lake water level (the coefficient of determination is equal to the square of the Pearson correlation coefficient, $r^2$, for linear regression), the $p$-value for the null hypothesis that the quantities are not related (Table 2), and the underlying data used to compute each $R^2$ and $p$ value (Fig. B3–B7). The coefficient of determination, $R^2$, is defined as the proportion of variance of a dependent variable (e.g., rate of discharge into moulins) predicted by an explanatory variable (e.g., surface melt rate). Quantities with a significant $p$-value ($p \lesssim 0.05$) can be interpreted as primarily

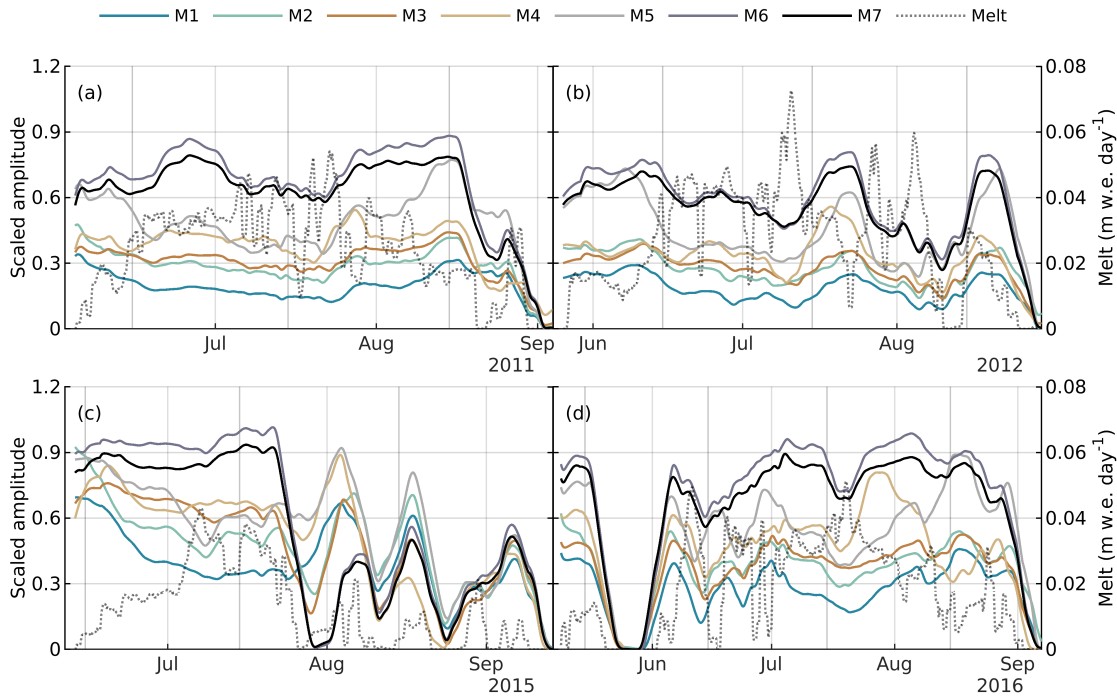

**Figure 3.** Relative diurnal amplitude of surface inputs to moulins (scaled by mean moulin input) and domain-averaged 24-hour rolling mean RACMO2.3p2 surface melt rate (right axis, dotted grey curve) for (a) 2011, (b) 2012, (c) 2015, and (d) 2016.

driven by surface melt, while quantities with a low $R^2$ and/or $p > 0.05$ are instead dominated by internal or other external variability in the system, including seasonal changes in the extent of supraglacial channels (Fig. B2). The $R^2$ and $p$-values are computed for direct model outputs at 2-hour resolution and for daily means. Table S1 provides complete statistics for all quantities, sub-catchments, and years.

Three of the five relationships considered in Table 2 are consistently significant across sub-catchments. Surface melt exerts the strongest control on the rate of inputs to moulins, with between 78% and 92% of the variance in daily mean moulin input explained by variations in diurnal-averaged melt rate ($p < 0.01$). The strength of the relationship does not vary significantly year-to-year (Fig. B3). When we analyze two-hour model outputs rather than the diurnal averages, the relationship remains statistically significant, but the proportion of the variance of moulin input rate explained by surface melt rate is lower for all catchments and all years (14–34%; $p < 0.01$; Table S1). The lower $R^2$ values obtained with two-hour model outputs may in part be due to the time lag between peak melt and peak moulin input rates.

The relationships between surface melt and relative diurnal amplitude, and surface melt and lag time, are not robust across all years and sub-catchments. Outside of specific examples (e.g., $R^2 = 0.62$ with $p < 0.01$ between surface melt rate and lag time for Moulin 3 in 2012; Fig. B5), variance in the amplitude and lag of moulin inputs is not significantly related to surface melt rate.

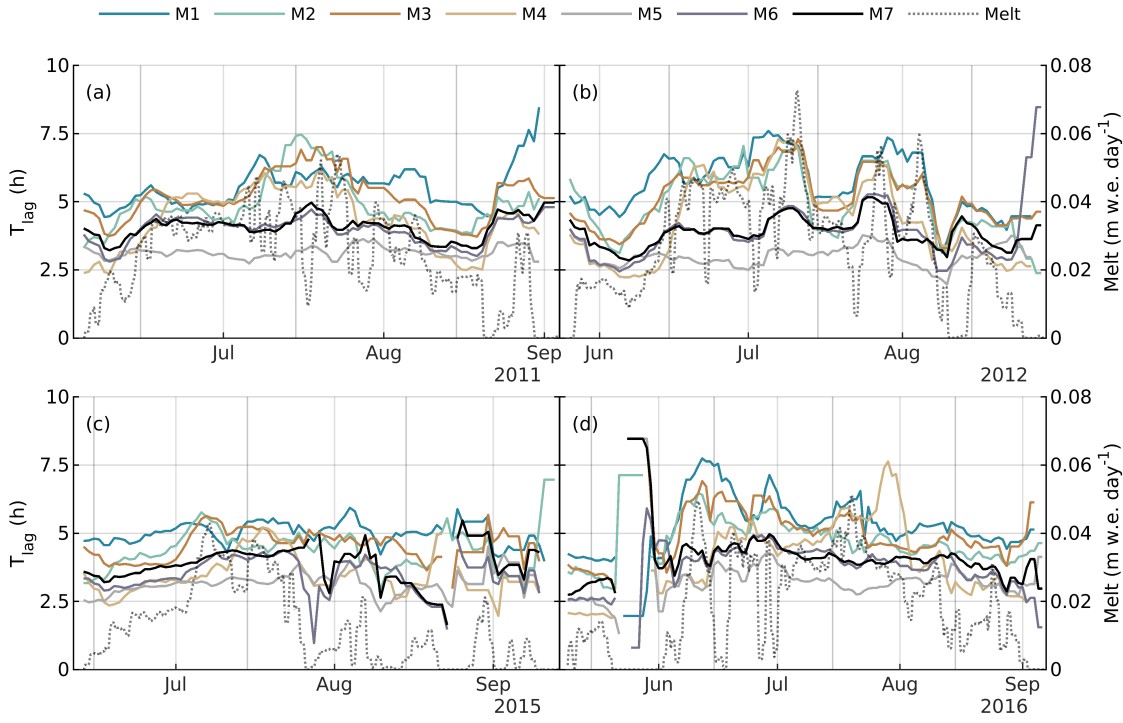

**Figure 4.** Seven-day moving-average time lag between local solar noon (15:22) and peak moulin input, and domain-averaged 24-hour rolling mean RACMO2.3p2 surface melt rate (right axis, dotted grey curve), for (a) 2011, (b) 2012, (c) 2015, and (d) 2016.

The water level in the four labelled supraglacial lakes is significantly controlled by melt rate, with some year-to-year variation. In 2012, 52–74% of variance in diurnal-averaged lake level for all four lakes is explained by diurnal-averaged melt rate ($p < 0.01$). In 2015, only 23% of variance in the daily mean water level in lake L3 is explained by surface melt rate. Melt rate exerts a weaker, yet still significant, control on lake level at the 2-hour timescale ($R^2 \leq 21\%$ for relationships with $p < 0.01$).

The relationship between lake level and the rate of input to the downstream moulin is significant in all years and all sub-catchments, at both daily and sub-daily timescales. Between 19–89% of variance in moulin input is explained by lake level at the sub-daily timescale, and 22–92% of variance in diurnal-averaged moulin input is explained by diurnal-averaged lake level with $p < 0.01$.

## 4   Discussion

### 4.1   Interannual variability in supraglacial drainage

The supraglacial drainage model predicts variations in the time lag between local solar noon and peak moulin inputs in response to interannual variations in melt forcing intensity and duration. For example, the time lag in 2015 (mean melt $1.13$ m w.e.)

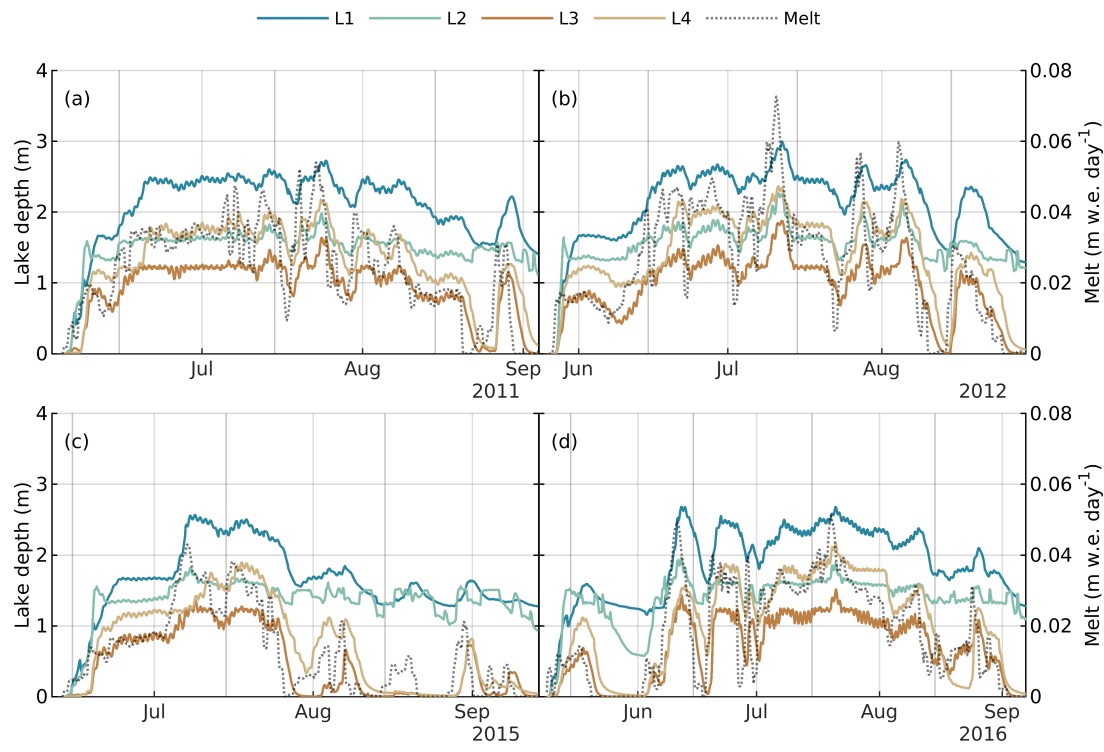

**Figure 5.** Supraglacial lake water depths and domain-averaged 24-hour rolling mean RACMO2.3p2 surface melt rate (right axis, dotted grey curve) for (a) 2011, (b) 2012, (c) 2015, and (d) 2016.

is typically below $\sim 5$ hours for all sub-catchments, while moulins M1–M4 reach as high as 7.5 hours in 2012 (mean melt 2.54 m w.e.). However, variations in the presented $R^2$ and $p$ values between surface melt rate and moulin input characteristics are more difficult to relate to surface melt volume. For example, compared to other sub-catchments, moulin M3 has the strongest

relationship between surface melt rate and time lag in 2012 and 2016, but not in 2011 and 2015 (when moulin M4 has the strongest relationship). The absolute coefficient of determination for M3 ranges between 0.62 in 2012 and 0.19 in 2015. These variations can not be entirely explained by total surface melt, since 2011 and 2016 have similar total melt volumes (2.19 m w.e. and 2.23 m w.e., respectively).

Interannual variations in surface melt rate have a clear impact on modelled lake water depth. Surface melt rate below

175 0.02 m w.e. day$^{-1}$ in August–September 2015, for example, leads to lower lake levels than during periods with melt rates exceeding 0.06 m w.e. day$^{-1}$ in 2012 (Fig. 5). The effect of these differences in lake water level on downstream moulins, however, appears to be basin-specific rather than uniform across basins. Lakes L1 and L2 show a reduced $R^2$ between lake water depth and moulin inputs in 2015, when water depth is the lowest, compared to 2011, 2012, and 2016. However, Lakes L3 and L4 exert the same influence on their downstream moulins in 2015 as they do in all other years (as measured by $R^2$ and

180 $p$ values).

**Table 2.** Coefficient of determination ($R^2$) and $p$-values for the null hypothesis that there is no relationship between the specified variables. Coefficients $R^2$ and $p$-values are computed independently for each of the seven sub-catchments and for each year. The tabulated min and max $R^2$ values represent the minimum and maximum $R^2$ values taken across the seven catchments for a given year, and the $p$ values represent the maximum value across the seven catchments. Coefficients $R^2$ and $p$-values are computed for model outputs at native 2-hour resolution and binned into 24-hour increments.

| Variables | | 2011 2-hour | 2011 24-hour | 2012 2-hour | 2012 24-hour | 2015 2-hour | 2015 24-hour | 2016 2-hour | 2016 24-hour |
|---|---|---|---|---|---|---|---|---|---|
| | N | 1105 | 93 | 1153 | 97 | 1129 | 94 | 1453 | 122 |
| Moulin input – | Min $R^2$ | 0.14 | 0.78 | 0.21 | 0.78 | 0.27 | 0.83 | 0.22 | 0.82 |
| surface melt | Max $R^2$ | 0.24 | 0.92 | 0.31 | 0.90 | 0.34 | 0.92 | 0.30 | 0.92 |
| | $p$ | $< 10^{-37}$ | <0.01 | $< 10^{-60}$ | $< 10^{-31}$ | $< 10^{-78}$ | $< 10^{-36}$ | $< 10^{-78}$ | $< 10^{-45}$ |
| Moulin input | Min $R^2$ | 0.00 | 0.01 | 0.00 | 0.00 | 0.01 | 0.01 | 0.00 | 0.00 |
| amplitude – | Max $R^2$ | 0.05 | 0.22 | 0.02 | 0.10 | 0.18 | 0.54 | 0.11 | 0.35 |
| surface melt | $p$ | 0.7 | 0.5 | 0.4 | 0.8 | 0.01 | 0.4 | 0.84 | 0.90 |
| Lag – surface | Min $R^2$ | 0.00 | 0.00 | 0.00 | 0.07 | 0.00 | 0.00 | 0.01 | 0.03 |
| melt | Max $R^2$ | 0.08 | 0.25 | 0.13 | 0.62 | 0.10 | 0.29 | 0.08 | 0.54 |
| | $p$ | 0.4 | 0.9 | 0.7 | 0.04 | 0.09 | 0.8 | 0.1 | 0.13 |
| Lake depth – | Min $R^2$ | 0.06 | 0.41 | 0.11 | 0.52 | 0.01 | 0.23 | 0.11 | 0.46 |
| surface melt | Max $R^2$ | 0.10 | 0.69 | 0.16 | 0.74 | 0.21 | 0.78 | 0.16 | 0.76 |
| | $p$ | $< 10^{-15}$ | $< 10^{-16}$ | $< 10^{-31}$ | $< 10^{-16}$ | 0.01 | $10^{-5}$ | 0.9 | $< 10^{-21}$ |
| Lake depth – | Min $R^2$ | 0.41 | 0.47 | 0.55 | 0.59 | 0.19 | 0.22 | 0.50 | 0.54 |
| moulin input | Max $R^2$ | 0.84 | 0.90 | 0.89 | 0.92 | 0.85 | 0.90 | 0.86 | 0.92 |
| | $p$ | $< 10^{-127}$ | $< 10^{-13}$ | $< 10^{-199}$ | $< 10^{-19}$ | $< 10^{-52}$ | $< 10^{-5}$ | $< 10^{-221}$ | $< 10^{-21}$ |

These seasonal changes in the behaviour of the supraglacial drainage system are more complicated than those shown in synthetic modelling. For example, Hill and Dow (2021) showed that increasing the intensity of surface melt applied to an idealized ice-sheet margin domain resulted in more clearly evident seasonal trends in the diurnal amplitude and lag time of surface inputs to moulins. The seasonal trends we observe here are instead convoluted with the highly variable surface melt rate signal over seasonal and annual timescales. It is possible that the highly variable surface melt rate signal could be separated from seasonal trends and interannual differences resulting from changes in the supraglacial drainage system by computing synthetic unit hydrographs (e.g., Smith et al., 2017) for each basin and each year, for example. However, this would be difficult with the limited three-hour temporal resolution climate model data used to force the supraglacial drainage model.

## 4.2 Seasonal trends in supraglacial drainage

Continuous seasonal trends in the amplitude and time lag of moulin inputs, as suggested by synthetic modelling (Yang et al., 2018; Hill and Dow, 2021), are not clear except in a few atypical cases. For example, in 2015, the diurnal amplitude of inputs

to moulins M1–M5 steadily decreases with a statistically significant trend ($p < 0.01$) from the onset of surface melting on 13 June until 2 July. Since this period (12 June to 2 July) is characterized by relatively steady surface melt rates ($\sim 1$ to $\sim 2$ cm w.e. day$^{-1}$), this trend may be a result of a reduction in the extent of small supraglacial channels (Fig. B2). The end of the decreasing trend coincides with a rapid increase in melt rate from $\sim 2$ to $> 4$ cm w.e. day$^{-1}$. Over the same period (12 June to 2 July), lag time is steady or slightly increasing aside from some transient behaviour in the first week of melting between 13 June and $\sim 19$ June (Fig. 3c, 4c).

Aside from limited examples of smooth, continuous seasonal trends, we observe discontinuous shifts in amplitude and time lag coinciding with rapid changes in surface melt rates. For example, in 2015 the relative diurnal amplitude of inputs to all seven moulins is tied to irregular melt intensities from $\sim 27$ July until the end of the melt season, $\sim 12$ September (Fig. 3c). The lack of continuous trends suggests that variations in surface melt rate are the primary control on moulin inputs compared to seasonal expansion and contraction of the supraglacial channel network (Fig. B2). The few examples of continuous trends (e.g., 13 June to 2 July 2015) occur during periods of steady surface melt rates, when the impact of the secondary controls (e.g., supraglacial channel evolution) are not obscured by melt rate variability.

## 4.3 Impact of supraglacial lakes

When they are present, supraglacial lakes influence the character of inputs to moulins. Large sub-catchments with lakes (M1–M4) have predominantly lower relative diurnal amplitude and longer lag times than the small catchments without lakes (M5–M7) (Fig. 4). The relationship between lake level and moulin inputs is evidenced by their correlation on diurnal ($0.22 \leq R^2 \leq 0.92$; $p < 0.01$) and sub-diurnal ($0.19 \leq R^2 \leq 0.89$; $p < 0.01$) timescales (Table 2). At least some of the remainder of the variance in moulin inputs is likely attributable to the fact that not all of the inputs to a moulin flow through the upstream lake, so that a proportion of the moulin inputs are unaffected by lake water level. The relationship between lake level and downstream moulin input is consistently stronger for lakes L3 and L4 (79–86%, $p < 0.01$ at daily scale), which do not have significant long-term storage, than for lakes L1 and L2 (19-77%; $p < 0.01$ at daily scale), which do store large volumes of water.

Compared to Lakes L1 and L2, there is negligible water depth stored in L3 at the end of each melt season (5), and despite similar input rates through M3 as M1 (Fig. 2), moulin M3 has consistently higher relative diurnal amplitude and a shorter lag time than M1 (Fig. 4, Fig. 5). Since lake L3 is also the only lake to fully drain following a pause in surface melt from 26–28 July 2015 (Fig. 5c), Lake L3 appears to behave more like a floodplain along the main channel through its sub-catchment than a lake with storage. This proposition is supported by the elongated shape of L3, lying directly along a main drainage channel, in the 8 July 2021 Landsat 8 scene (Fig. 1a).

Given the position of M4 directly within the L4 basin (this position explains the lack of long-term storage in L4), it is possible that L4 rapidly drains directly through M4, in contrast to the other lakes in the domain which could drain through unstable incision of their outlet channel (Kingslake et al., 2015). Satellite imagery from 8 July 2021 does not show a lake in the location of L4, despite visible lakes upstream from our study site at higher elevations (Fig. 1a). Unfortunately, the dataset

used to derive moulin locations (Yang and Smith, 2016) does not discriminate between stable and rapidly draining lakes, so we do not know if this is the case.

    The behaviour of lakes L1 and L2 is further differentiated from that of L3 and L4 by their control on downstream moulins in the low melt year (2015). For 2011, 2012, and 2016 (mean melt $\geq 2.20$ m we.), the water level in L1 and L2 explains at least 47% (L1–M1) and 74% (L2–M2) of the variance of downstream moulin inputs. In 2015 (mean melt 1.13 m we., these

lakes explain only 23% (L1–M1) and 49% (L2–M2) of the variance of downstream moulin inputs. The L3–M3 and L4–M4 relationships are not affected by low surface melt rates in 2015.

    Aside from the combined influence of domain size and supraglacial lakes in modulating the amplitude and time lag of moulin inputs, there is no significant difference in the strength or significance of the relationship between surface melt and moulin inputs, the amplitude of moulin inputs, or their timing between catchments with and without lakes (Table 2, S1).

An unfortunate limitation of our model domain is that we can not completely disentangle the effects of catchment size and supraglacial lakes, since the four labelled lakes occupy the four largest catchments, so it is not possible to determine exactly the contrasting drivers of moulin input dynamics in catchments with and without lakes.

### 4.4    Streamflow observations

It remains difficult to constrain supraglacial drainage models such as SaDS due to a limited number of sufficiently long in-situ

discharge records. We require a long (ideally entire melt-season) record since one of the primary advantages of SaDS is that it dynamically represents seasonal changes in the supraglacial drainage system efficiency. Muthyala et al. (2022) present one of the only sufficiently long continuous records (62 days from 2016) from a small (0.6 km$^2$) lake-free catchment in southwest Greenland. Within our study site, catchments M6 and M7 are the most similar to the catchment instrumented by Muthyala et al. (2022). A detailed quantitative comparison by applying SaDS to the same catchment is not presently possible without

complete in-situ data, since it would not be possible to separate discrepancy arising from differences in surface melt forcing and DEM surface elevations from discrepancy arising from model error. Despite this limitation, we can compare moulin inputs qualitatively.

    For M6 and M7 (combined catchment area of 8.5 km$^2$), we have modelled lag times between 2–5 hours, compared to 1–3 hours from Muthyala et al. (2022). SaDS predicts a relative diurnal amplitude of 50–100%, i.e., flow nearly pauses overnight,

in qualitative agreement with data collected by Muthyala et al. (2022) showing stream discharge minimums near zero except during periods of elevated positive overnight energy balance. Muthyala et al. (2022) report similar abrupt shifts in the timing and amplitude of streamflow related to rapid changes in melt forcing as are observed from the SaDS outputs (Fig. 2, 3, 4). Muthyala et al. (2022) also report a statistically significant decrease in the lag time between solar noon and peak streamflow from 25 June until 4 August 2016, terminating when daily minimum air temperatures decreased from $\sim 3°C$ to $\sim 1°C$. We

do not capture similar trends in lag time in our model outputs, however we have identified periods of continuous trends in the diurnal amplitude of moulin inputs that terminate with a rapid shift in surface melt rates. Since we have observed that melt rate variations mask seasonal trends, it remains possible that we would observe such a trend in lag time with a different melt forcing timeseries. Despite these specific details, the streamflow measurements (Muthyala et al., 2022) demonstrate a similar

pattern of a melt rate-dominated system, with internal dynamics (e.g., changes in the extent of supraglacial streams, filling and drainage of supraglacial lakes) as a secondary control that becomes important during periods of steady surface melt rates.

It is also possible to compare our model results to shorter measurement campaigns, however these comparisons do not constrain the seasonal evolution of supraglacial behaviour. For the $\sim$60 km$^2$ Rio Behar catchment (67.047°N, -49.033°W) with flow <40 m$^3$ s$^{-1}$ in the main river stem leading to the terminal moulin, Smith et al. (2017) report a lag time of 5.5 hours from Acoustic Doppler Current Profiler (ADCP) measurements from 20 to 23 July 2015, and Smith et al. (2021) report a lag time of 6 hours from repeated ADCP measurements on the same river segment from 6 to 13 July 2016. Our modelled inputs for moulins M1 and M3 ($\lesssim$40 m$^3$ s$^{-1}$ in 2011, 2015, and 2016) have similar input rates to the Rio Behar measurements and show a similar lag time ($\sim$4–7 hours). Lag times for M1 and M3 are higher in 2012 ($\sim$5–7 hours) when the absolute magnitude of moulin inputs is also much larger (up to >60 m$^3$ s$^{-1}$). However, these differences in lag time should be interpreted with caution since Smith et al. (2017) and Smith et al. (2021) report lag times relative to peak melt rather than solar noon. This difference could be important, for example, if local weather conditions modulate the timing of peak melt relative to solar noon (e.g., Smith et al., 2021). On the other hand, the difference in timing between solar noon and peak melt reported by Mejia et al. (2022) for $0.2\,\mathrm{km}^2$ and $16.7\,\mathrm{km}^2$ catchments is less than three hours, so we would not be able to resolve these differences with our three-hour resolution surface melt forcing data.

These comparisons suggest that SaDS predicts reasonable lag times and amplitudes compared to observations with similar discharges. However, given the currently available observational data, it remains difficult to calibrate the model and determine the most relevant mechanisms by which supraglacial drainage evolves. A full calibration dataset would ideally contain streamflow observations spanning multiple orders of discharge magnitude, measurements of lake water level, and the evolution of channel cross sections, for a significant portion of the melt season.

### 4.5 Model intercomparison and selection

SaDS is among the models with the most complete supraglacial hydrologic process representation, including both channelized and distributed drainage, filling and draining of supraglacial lakes, and dynamically computed seasonal evolution of supraglacial stream density. Compared to flow-routing type models (e.g. Banwell et al., 2012; Yang et al., 2018), travel times in SaDS depend on upstream flow accumulation and melt rate, so that the time lag and relative diurnal amplitude of moulin inputs varies in a complex way with melt forcing.

Modelled water levels in supraglacial lakes fluctuate by several centimeters in response to diurnal melt variations and by up to a meter over a few days in response to larger changes in melt intensity. It is not clear how this variation is partitioned between nonlocal processes such as downstream channel capacity adjustment and local processes such as excess meltwater transiently being stored in lakes. These rapid fluctuations in lake level may partly explain the residual variance in the timing of rapid lake drainage not explained by strain rates (Poinar and Andrews, 2021). However, these fluctuations may not be captured by models treating lakes as holding a prescribed storage volume (e.g. Banwell et al., 2012).

An incised channel network (c.f. Leeson et al., 2012) is important for efficiently routing large volumes of meltwater into moulins (Yang et al., 2018; Gleason et al., 2021). Channel density may also change through the melt season, changing the

response of moulin inputs to melt forcing (Yang et al., 2022). Compared to these models, SaDS dynamically evolves channel density based on the local balance between incision and melt-out for each channel element.

The process-based approach taken by SaDS increases the computational cost well over that of other models ($\sim$7 days per year of simulation time in the configuration presented here). This tradeoff between physics and computational cost yields a spectrum of models from which the appropriate model must be chosen for each application so as not to neglect key features of supraglacial drainage.

We recommend choosing the appropriate supraglacial drainage model based on the sensitivity of a prospective modelling study to the supraglacial drainage characteristics that we have highlighted here. If short-term (<1 day) variations in the rate, amplitude, and time lag of moulin inputs will materially change the results of the modelling study, for example when modelling the short-timescale variations in moulin water storage (e.g. Andrews et al., 2022), it may be important to use a detailed process-resolving model such as SaDS. If only aggregate quantities, e.g. the total volume of moulin inputs on seasonal timescales for assessing centennial-scale changes in ice sheet subglacial hydrology, are relevant, it should be possible to carefully select a less expensive model that captures moulin inputs with an appropriate level of detail (e.g., Banwell et al., 2012; Leeson et al., 2012; Yang et al., 2018). The more difficult case is where day-to-day variation in the character of moulin inputs will impact the modelling study results in a noticeable way, and where the computational cost of a process-resolving supraglacial hydrology model will be a rate-limiting step in the modelling workflow. For example, this may be the case when modelling subglacial drainage on seasonal timescales. Here, the sensitivity of the outcomes of the modelling study (e.g., diurnal and seasonal variations in subglacial effective pressure) to the character of moulin inputs could be quantified by carrying out benchmark simulations with simpler models (e.g., Yang et al., 2018) with varying parameters. The experimenter can then make an informed decision on the appropriate supraglacial drainage model based on the known sensitivity. For example, it would be valuable to combine process-resolving supraglacial (e.g., SaDS), englacial (Andrews et al., 2022), and subglacial (e.g., GlaDS; Werder et al., 2013) models to quantify the sensitivity of subglacial drainage to supraglacial and englacial processes.

## 5 Conclusions

We have presented model outputs of supraglacial meltwater flow, lake filling and draining, and surface inputs to moulins for a small ($\sim$20 $\times$ 27 km$^2$) catchment on the Greenland Ice Sheet for four years, including the extreme melt year in 2012. The relationship between surface melt and the magnitude, relative diurnal amplitude, and timing of surface inputs to moulins is complex and sensitively depends on the duration and intensity of surface melt.

Supraglacial lakes exert a strong control on moulin inputs. The four large catchments with supraglacial lakes within our domain have consistently lower relative diurnal amplitude in moulin inputs (however, a larger absolute diurnal amplitude given the larger magnitude of moulin inputs) and a longer time lag between solar noon and peak moulin input than the three small catchments without lakes. However, since we find lakes only in the largest catchments, the relative contributions of catchment scale and lakes can not be fully disentangled with the current model outputs.

SaDS model outputs contain similar key features (time lag and diurnal amplitude) to in-situ streamflow observations from a nearby catchment (Muthyala et al., 2022). However, given the two orders of magnitude spanned by moulin inputs within our study site, we can not yet observationally constrain SaDS without additional long-term stream discharge records spanning a wide range of discharge magnitudes and for catchments with and without lakes.

By placing these results in the context of existing drainage models with differing levels of process representation and com-
330 putational expense, we have suggested a conceptual model for selecting the appropriate drainage model for a prospective modelling study. The supraglacial model should be chosen based on expected or demonstrated sensitivity to the exact nature of moulin inputs. A process-resolving model such as SaDS is expected to be advantageous when variations in the amplitude and timing of moulin inputs on sub-daily timescales will substantially change the outcomes of the prospective modelling study. When only a broad representation of moulin inputs is required, simpler routing models are expected to be appropriate and
335 come with a substantially reduced computational cost.

## Appendix A:  Model description

SaDS follows the general structure of the GlaDS model (Werder et al., 2013). The model is posed on an unstructured triangular mesh with distributed sheet-like flow across elements and discrete channelized flow along edges. The key dynamic equations are reviewed here; for full details refer to Hill and Dow (2021).

### 340 A1 Distributed drainage

Distributed flow across the ice surface is governed by conservation of mass, written in terms of the flow depth $h_\mathrm{s}$,

$$\frac{\partial h_\mathrm{s}}{\partial t} + \nabla \cdot \mathbf{q}_\mathrm{s} = m_\mathrm{s} + \frac{\Xi_\mathrm{s}}{\rho_\mathrm{w} L} - \frac{m_e}{l}, \tag{A1}$$

for discharge per unit width $\mathbf{q}_\mathrm{s}$, distributed melt rate $m_\mathrm{s}$, and potential energy dissipation $\Xi_\mathrm{s}$. The final term represents the rate of mass transfer $m_e$ into a channel with length $l$. Constants are density of water $\rho_\mathrm{w}$ and latent heat of fusion $L$. The discharge
$\mathbf{q}_\mathrm{s}$ is approximated with a Darcy-Weisbach-like equation,

$$\mathbf{q}_\mathrm{s} = -k_\mathrm{s} h_\mathrm{s}^{\alpha_\mathrm{s}} \left| \frac{\nabla \phi_\mathrm{s}}{\rho_\mathrm{w} g} \right|^{\beta_\mathrm{s}-2} \frac{\nabla \phi_\mathrm{s}}{\rho_\mathrm{w} g}, \tag{A2}$$

for hydrostatic potential $\phi_\mathrm{s}$, conductivity $k_\mathrm{s}$, and gravity $g$. Exponents $\alpha_\mathrm{s}$ and $\beta_\mathrm{s}$ control the form of the equivalent Darcy friction factor.

### A2 Channelized drainage

Supraglacial channels are parameterized by the total incision depth of the channel $H_\mathrm{c}$ and the depth of water flowing in the channel $h_\mathrm{c}$. The channel width is assumed to be related to channel depth by $w_\mathrm{c} = r H_\mathrm{c}$ for a constant and uniform width-to-depth ratio $r$. The dynamic evolution of channel incision depth is governed by the balance between melt along the channel bed

and walls by potential energy dissipation and melt-out of the adjacent ice surface,

$$\frac{dH_c}{dt} = \frac{1}{2}\frac{\rho_w}{\rho_i}(m_c - m_s) + \frac{1}{2w_c}\frac{\Xi_c}{\rho_i L}, \tag{A3}$$

for channel bed melt rate $m_c$, sheet melt rate $m_s$, potential energy dissipation $\Xi_c$, and ice density $\rho_i$.

The dynamic evolution of channel flow depth is given by conservation of mass,

$$w_c\frac{\partial h_c}{\partial t} + \frac{\partial q_c}{\partial t} = w_c m_c + \frac{\Xi_c}{\rho_w L} - h_c\frac{\partial w_c}{\partial t} + m_e, \tag{A4}$$

where the channel discharge $q_c$ is computed with an analogous expression to (A2) using the channel hydrostatic potential, $\phi_c = \rho_w g z_c + \rho_w g(h_c - H_c)$, for channel lip elevation $z_c$. $m_e$ represents mass transfer from the distributed system into the
channel.

## A3    Drainage system coupling

Unlike GlaDS, SaDS uses different hydraulic potentials in distributed ($\phi_s$) and channelized ($\phi_c$) systems, so SaDS requires an explicit rule to exchange mass between the systems. The mass exchange over an edge of length $l$ is computed as

$$m_e = flq_n, \tag{A5}$$

for exchange fraction $f$ and normal flux $q_n = \mathbf{q}_s \cdot \mathbf{n}$ where $\mathbf{n}$ is the outward unit normal of the edge. The exchange fraction $f$ depends on the difference in potential between the channel and adjacent sheets such that mass is only transferred into the channel when the channel potential is less than the sheet potential.

**Table A1.** SaDS model parameters.

| Parameter | Description | Value | Units |
|---|---|---|---|
| $\alpha_s$ | Distributed sheet flow exponent | $\frac{5}{4}$ | - |
| $\beta_s$ | Distributed sheet flow exponent | $\frac{3}{2}$ | - |
| $\alpha_c$ | Channel flow exponent | $\frac{5}{3}$ | - |
| $\beta_c$ | Channel flow exponent | $\frac{3}{2}$ | - |
| $k_s$ | Distributed sheet hydraulic conductivity | 1.0 | $\mathrm{m}^{(2-\alpha_s)}\,\mathrm{s}^{-1}$ |
| $k_c$ | Channel hydraulic conductivity | 15 | $\mathrm{m}^{(2-\alpha_c)}\,\mathrm{s}^{-1}$ |
| $r$ | Channel width-to-depth ratio | 5.0 | - |
| $\zeta$ | Exchange ratio | 0.2 | - |
| dt | Timestep | 20 | s |
| $\rho_i$ | Density of ice | 910 | $\mathrm{kg\,m}^{-3}$ |

## Appendix B: Extended figures

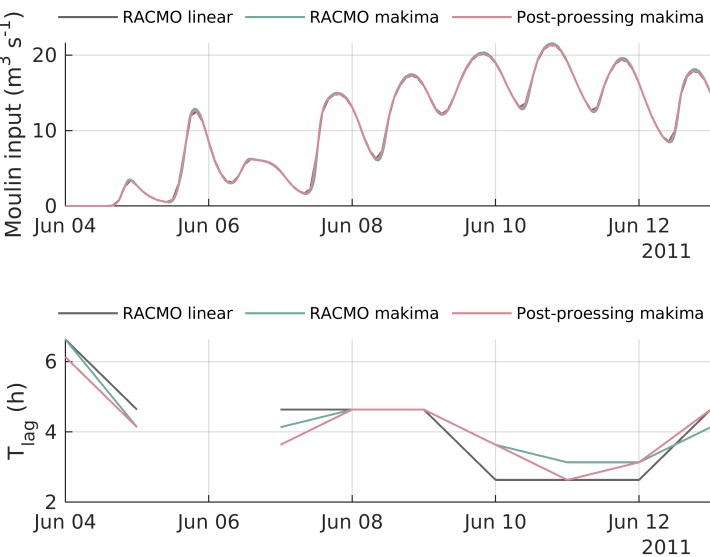

**Figure B1.** Comparison of the rate of surface inputs to moulins (a) and the time lag between local solar noon and peak moulin inputs (b) for the beginning of the 2011 melt season for three interpolation schemes. (1) 'RACMO linear' uses linear interpolation to downscale RACMO from native 3 hour resolution to the 20 s model timestep and no post-processing interpolation to compute time lag. (2) 'RACMO makima' uses modified Akima interpolation to downscale RACMO from 3 hour to 20 s timestep and no post-processing interpolation to compute time lag. (3) 'Post-processing makima' uses linear interpolation to run the model, and interpolates model outputs onto 30 min timesteps using modified Akima interpolation before computing lag time.

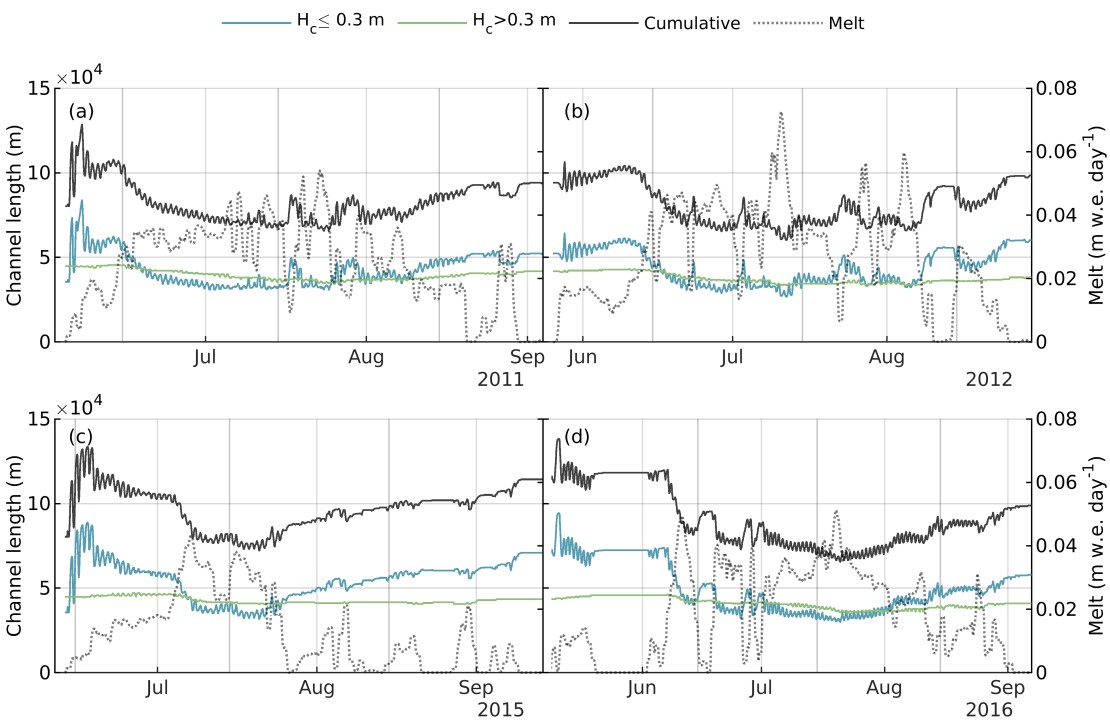

**Figure B2.** Total length of incised supraglacial channel network, partitioned between small (incision depth less than or equal to 0.3 m) and large (incision depth greater than 0.3 m) channels, for (a) 2011, (b) 2012, (c) 2015, (d) 2016, and domain-averaged 24-hour rolling mean RACMO2.3p2 surface melt rate (right axis, dotted grey curve).

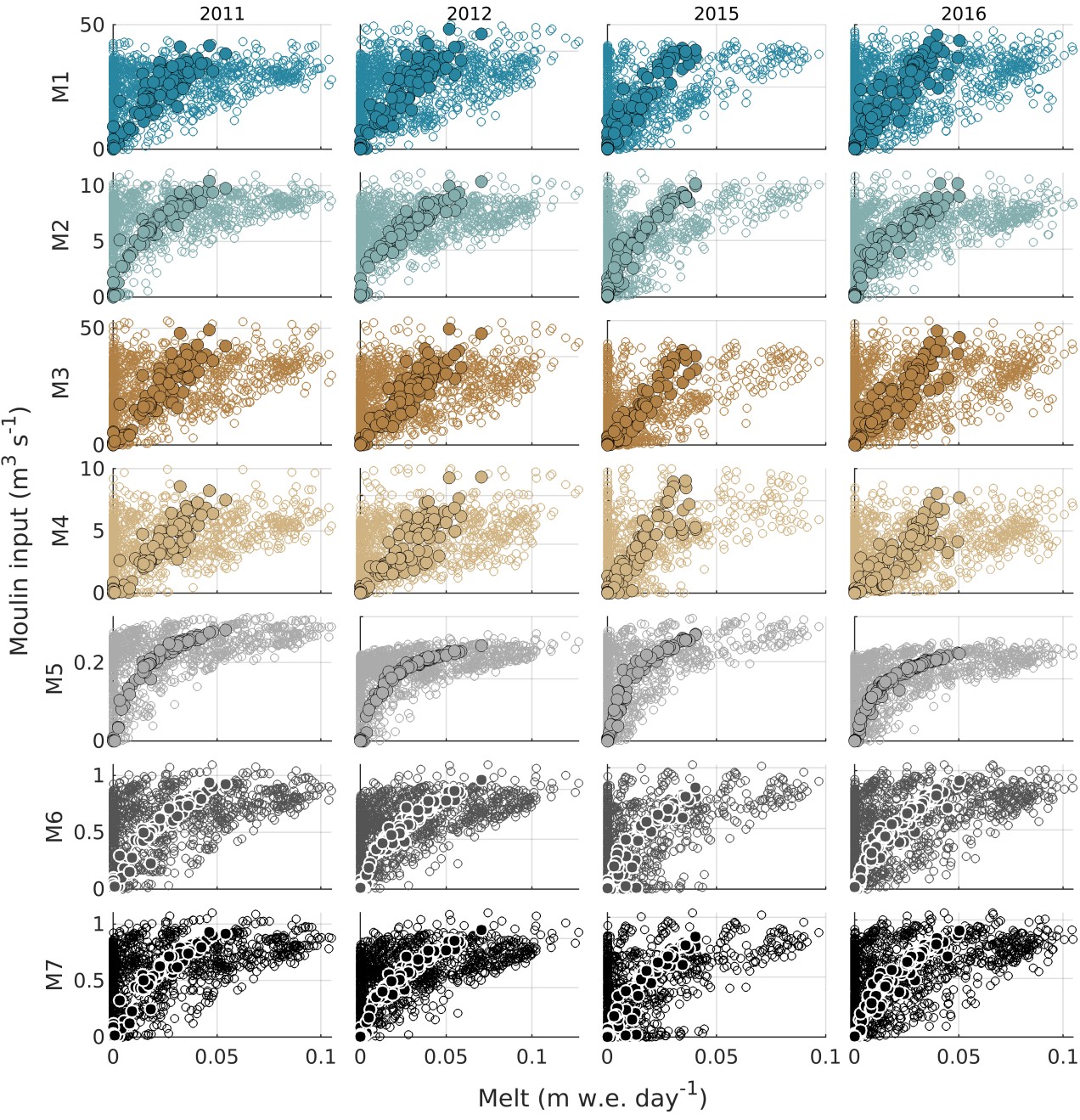

**Figure B3.** Relationship between surface melt rate and moulin inputs for sub-catchments M1–M7 and for 2011, 2012, 2015, and 2016. Hollow markers represent modelled moulin inputs at native two-hour resolution and filled markers with black or white outlines represent the 24-hour moving average melt rate and moulin input.

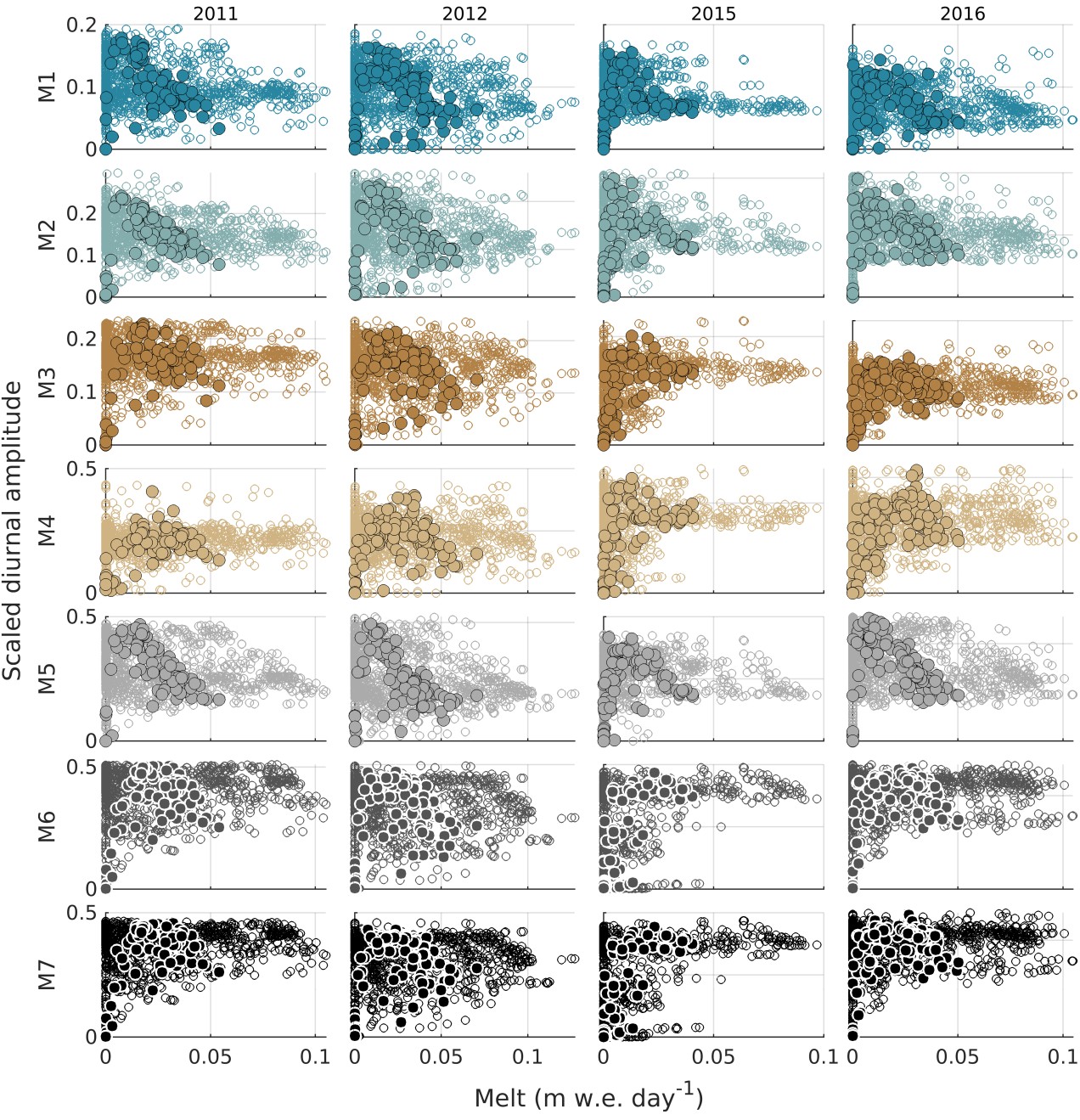

**Figure B4.** Relationship between surface melt rate and relative diurnal amplitude for sub-catchments M1–M7 and for 2011, 2012, 2015, and 2016. Hollow markers represent modelled diurnal amplitude interpolated to native two-hour resolution and filled markers with black or white outlines represent the daily relative diurnal amplitude.

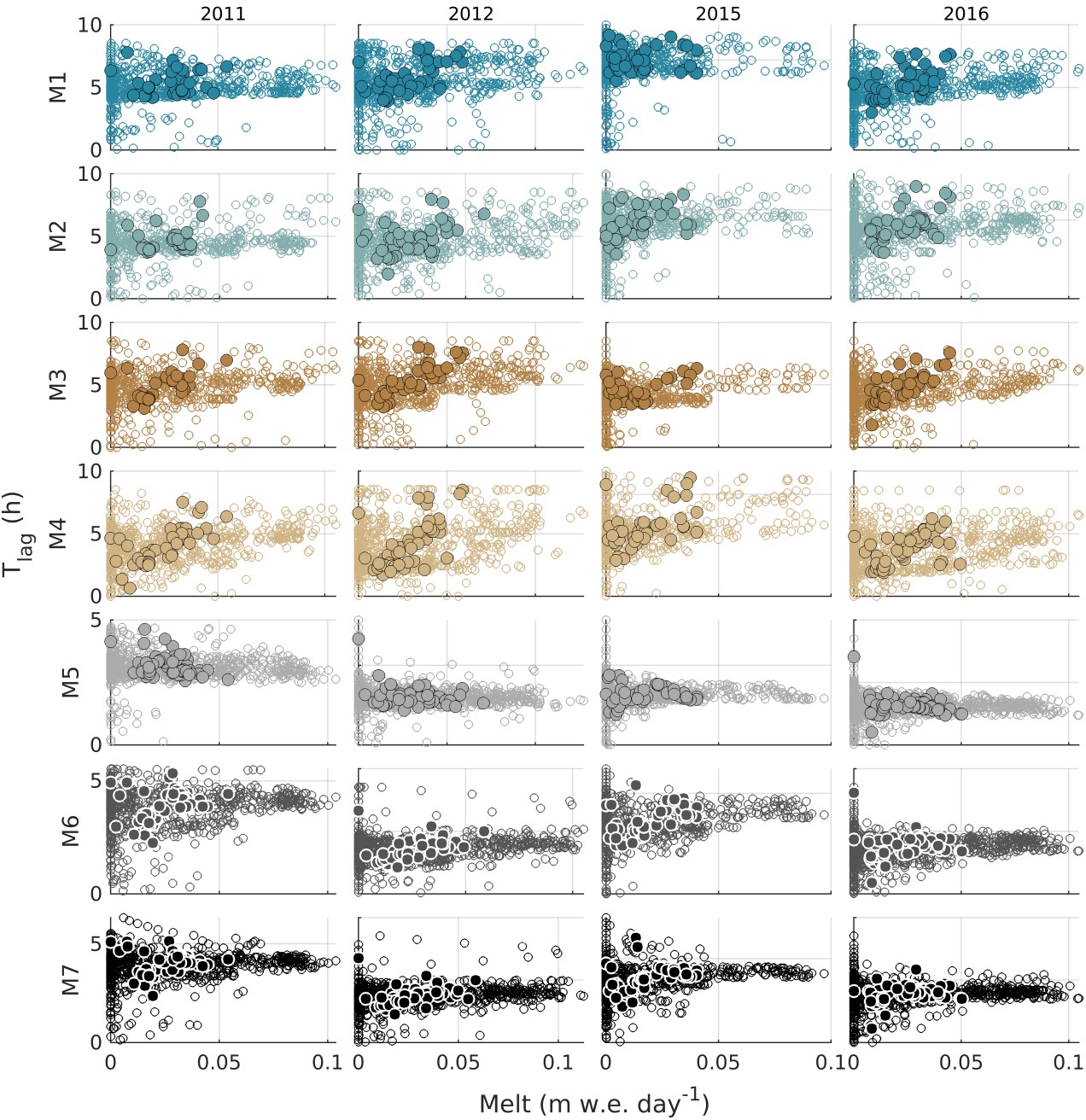

**Figure B5.** Relationship between surface melt rate and the time lag from local solar noon to peak moulin discharge for sub-catchments M1–M7 and for 2011, 2012, 2015, and 2016. Hollow markers represent modelled lag time interpolated to native two-hour resolution and filled markers with black or white outlines represent the daily lag time.

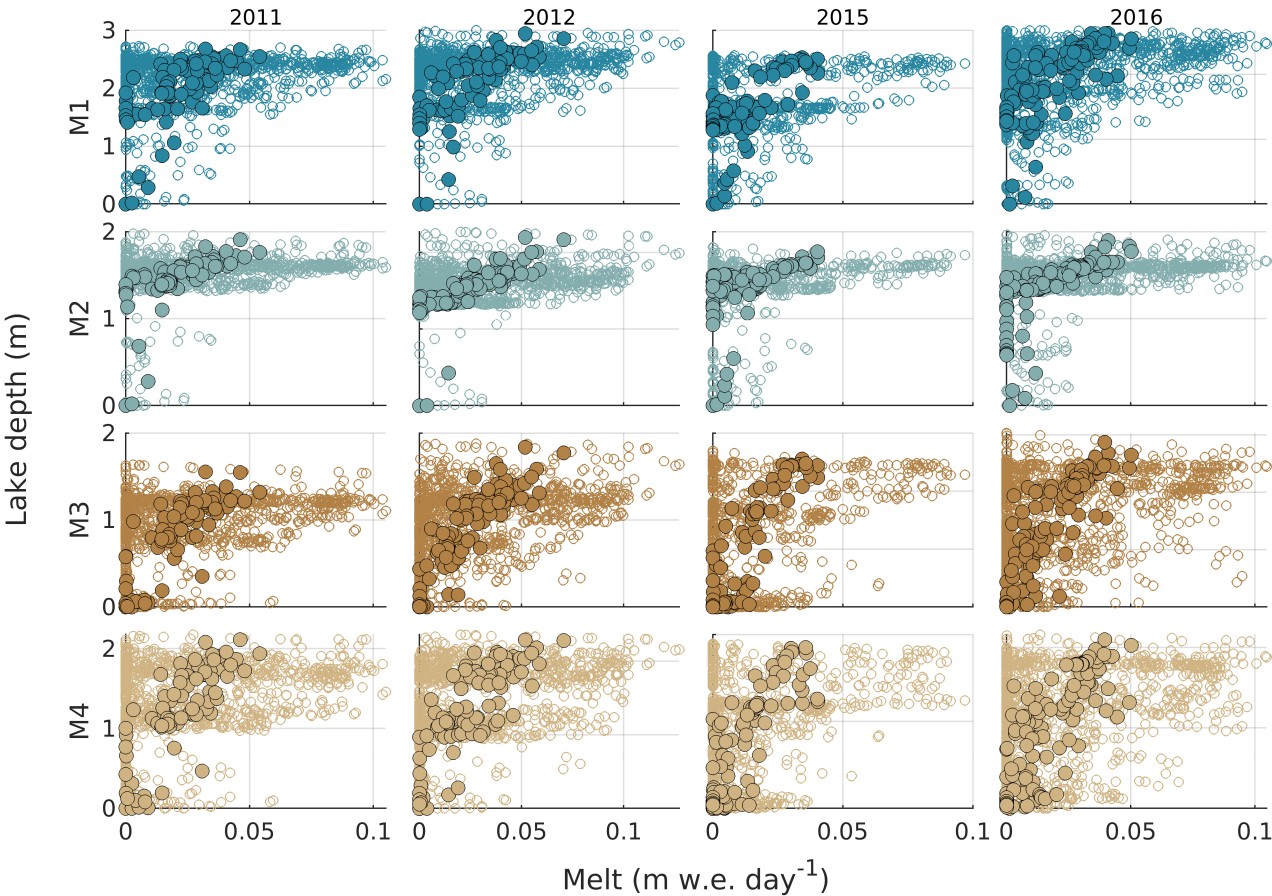

**Figure B6.** Relationship between surface melt rate and lake water depth for lakes L1–L4 and for 2011, 2012, 2015, and 2016. Hollow markers represent modelled lake water depth at native two-hour resolution and filled markers with black outlines represent the 24-hour moving average melt rate and lake depth.

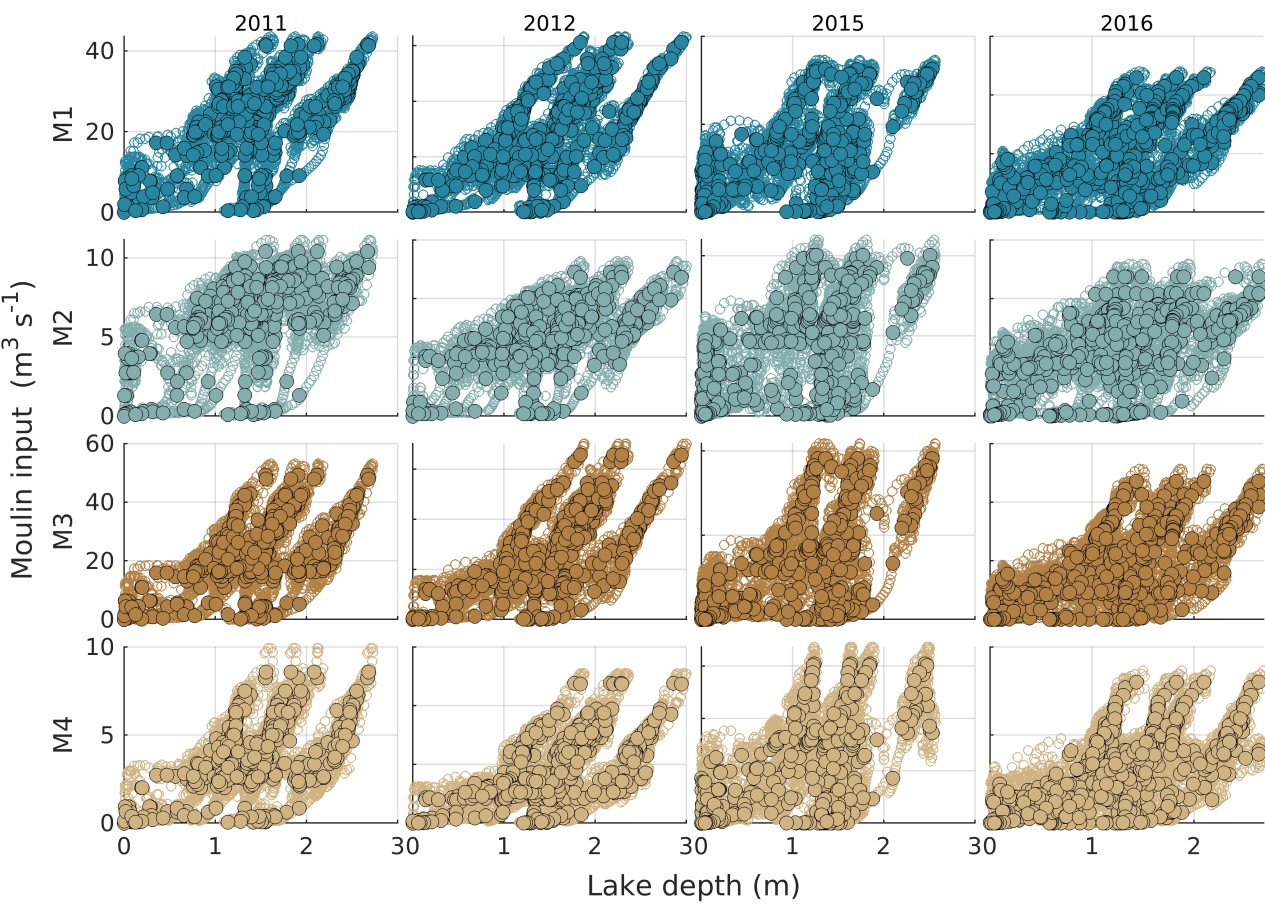

**Figure B7.** Relationship between lake water depth and moulin input for sub-catchments L1–L4 and for 2011, 2012, 2015, and 2016. Hollow markers represent modelled moulin inputs at native two-hour resolution and filled markers with black or white outlines represent the 24-hour moving average lake depth and moulin input.

*Code and data availability.*  ArcticDEM data are freely available from the Polar Geospatial Center (https://www.pgc.umn.edu/data/arcticdem/).
RACMO surface melt data are available by contacting the Institute for Marine and Atmospheric research Utrecht University (https://www.
projects.science.uu.nl/iceclimate/models/racmo.php). The SaDS model is described fully in Hill and Dow (2021) including equations and
implementation. Table S1 and all model outputs and code to produce the figures are available at https://doi.org/10.5281/zenodo.7968634.

*Author contributions.*  T Hill contributed to conceptualization, analysis, methodology, software, visualization, writing and reviewing. C Dow
contributed to conceptualization, analysis, methodology, supervision, and reviewing & editing.

*Competing interests.*  The authors declare that they have no conflict of interest.

*Acknowledgements.*  T Hill was supported by the Ontario Graduate Scholarship and NSERC Canada Graduate Scholarship programs. C Dow
was supported by the Natural Sciences and Engineering Research Council of Canada (RGPIN-03761-2017) and the Canada Research Chairs
Program (950-231237). The authors declare that they have no conflicts of interest. DEM provided by the Polar Geospatial Center under
NSF-OPP awards 1043681, 1559691, and 1542736. The authors would like to thank Brice Noël and Michiel van den Broeke for providing
RACMO surface melt data.

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
