# Peer review of "The impact of surface melt rate and catchment characteristics on Greenland Ice Sheet moulin inputs"

_The Cryosphere, 2022_

## Author Response (AR1)

**Author Response for "The impact of surface melt rate and catchment characteristics on Greenland Ice Sheet moulin inputs"**

Tim Hill & Christine F. Dow

| Reviewer Comment | Author Response | Manuscript changes |
|---|---|---|

**Reviewer 1**

**General Comments**
In this paper the authors apply their Subaerial Drainage System (SaDS) model (Hill and Dow 2021), a supraglacial meltwater routing model, to a ~20 km$^2$ area for four melt seasons to assess how the supraglacial drainage system and moulin discharge changes between years with high and low melt intensities. At present there is not a description of the SaDS model in the manuscript or in the supplement which prevents the readers from understanding the physics of this approach without referring to the other manuscript. Further, there are no comparisons of the model to any observations over the four years considered here, without any description of model validation it is hard to determine if the results are physically meaningful. At present the paper's main findings are unclear. I do see potential in the work presented here, however, major revisions are required to address the structural and clarity issues and further develop the ideas presented for readers to understand or have confidence in results and conclusions derived from this work.

We appreciate the reviewer's comments and suggestions. The concerns raised here are addressed individually below, followed by a high level plan to address both reviewer's comments about structure and clarity (**General Changes**).

We have implemented the plan described in **General Changes** to address the comments by both reviewers.

**Specific Comments**
1. The manuscript currently lacks any real description of the model, a more thorough description needs to be included in the main text with supporting details in the supplement. The results and discussion mention several model parameters which are never introduced, explained or justified (e.g., sheet mass, channel mass, lake depth, and incised channel length/channel incision depth with no figures that correspond to the details stated in the results).

The current manuscript intentionally did not fully describe the model in terms of its governing equations and parameters since this description is presented in detail in Hill & Dow (2021). However, to address this concern we have written an Appendix summarizing the key model equations, and we will describe the relevant mechanisms in the Introduction.

Specific model parameters (Table S2) are not discussed in the main text. We believe the descriptions of model outputs (e.g., sheet mass, channel mass, and their associated units etc.) are sufficiently accessible that they do not need to be described in detail with reference to the relevant governing equations from Hill & Dow (2021). If necessary, readers will be able to refer to the appendix for more details.

Model equations and a more complete model description have been added to Appendix A, and the description of the model in Lines 34 to 37 and 54 to 64 has been expanded.

2. The structure of the manuscript needs some reworking. For example a majority of the paper is framed to focus on the timing of meltwater delivery to moulins and how this evolves over the melt season, however, a majority of the discussion focuses on supraglacial lakes, introducing Figure 3 that is not included in the Results section or described in the methods, and then does not mention lakes again in the conclusions. Similarly the end of the discussion compares the SaDS model to other modeling works that are not mentioned in the introduction, much of this content should be moved to the introduction to explain why this model is being used, what makes it unique, and how it differs from other models.

We agree that the focus of the paper was inconsistent between sections. See the **General Changes** section below for our proposal to restructure the manuscript.

We have implemented these changes to focus on the relationship between seasonal variations in surface melt volume, compare SaDS to available drainage models, and provide guidance on when it may be appropriate to use a model such as SaDS.

3. There are many ideas and concepts that are either not introduced and come out of nowhere or that are briefly discussed but never resolved. (i) The paper focuses on seasonal trends in drainage system behavior however Section 4.1 is very brief and makes several statements without a robust discussion. There are several statements that are unsupported such as that decreasing moulin diurnal amplitude is caused by the extent of the supraglacial channel network, however, there are no figures that show supraglacial channel network evolution.
As part of the **General Changes** we plan to make, we will remove unsupported statements (e.g., decreasing moulin diurnal amplitude is caused by changes in the channel network), and instead focus on the lack of smooth, continuous trends in amplitude and time lag in most years. As with Muthyala et al. (2022), we observe distinct changes in lag time and diurnal amplitude related to changes in melt rate.

The discussion has been refocused to evaluate specific features observed in our model outputs and compare these features to those observed by Muthyala et al. (2022), Smith et al. (2017), and Smith et al. (2021).

What are the errors associated with your lag times and the ones presented by Muthyala et al? How does this factor in to the comparison you present?

Based on tests with different interpolation methods, the error in lag time related to the linear interpolation scheme applied to RACMO runoff data is small (0–0.5 hours). See also our response to the L65–68 comment. The limitations of using RACMO runoff outputs are one reason we have not applied SaDS to the domain of Muthyala et al. (2022) for a direct comparison. Without in-situ melt forcing data at sufficiently high temporal and spatial resolution, it would be difficult to disentangle discrepancy arising from incorrect melt forcing from discrepancy related to the SaDS model. However, given the lack of in situ melt data available for much of Greenland, we believe our approach using RACMO is justified in the broader modelling context. We now discuss the limitations of our linear interpolation scheme and more generally the limitations of using RACMO data in the manuscript in section 2.3.

We have addressed the inability to apply SaDS directly to the domain of Muthyala et al. (2022) in Lines 220 to 223.

> A detailed quantitative comparison by applying SaDS to the same catchment is not presently possible without complete in-situ data, since it would not be possible to separate discrepancy arising from differences in surface melt forcing and DEM surface elevations from discrepancy arising from model error. Despite this limitation, we can compare moulin inputs qualitatively.

Also, this section states that the lack of a trend is because of the domain size and number of supraglacial lakes, how these two things would affect moulin discharge and diurnal amplitudes should be discussed and cited.

We will update this comparison to include the two most similar sub-catchments within our domain, which have similar stream discharge (< 1 $m^3$ $s^{-1}$) and do not contain supraglacial lakes to remove the unsupported statements related to catchment size and the influence of lakes. We have found that the smaller catchments without supraglacial lakes result in moulin inputs with higher diurnal amplitudes and shorter lag times.

We have compared our results for the two most similar sub-catchments to Muthyala et al. (2022). The main findings are that the modelled diurnal amplitude from SaDS is within the observational range, and that modelled lag times are slightly longer (2–5 hours from SaDS, 1–3 hours from Muthyala et al., 2022) (Line 224 to 236).

Further, this section puts this work in the context of only one other paper Muthyala et al., 2022, and does not mention other studies (e.g., Yang and Smith 2016, Yang et al., 2018, Smith et al., 2021). It would be interesting to know how these model results relate to the wider supraglacial drainage system literature.

The model results are compared to the wider literature in Section 4.4. In this section we only compare to Muthyala et al. (2022) since it is the only study with a sufficiently long discharge timeseries to observe seasonal trends in discharge amplitude and timing. Smith et al. (2017) and Smith et al. (2021), for example, only present discharge records for 3–7 days and we are primarily interested in differences between years. We will explain this reasoning in the

manuscript. The SaDS outputs were compared against Yang et al. (2018) in Hill and Dow (2021) and we refer readers to this in the manuscript.

We have added a comparison of modelled lag times between SaDS and Smith et al. (2017) and Smith et al. (2021), however the comparison is limited due to the short duration of these stream discharge records (Line 237 to 244).

> It is also possible to compare our model results to shorter measurement campaigns, however these comparisons do not constrain the seasonal evolution of supraglacial behaviour. For the ~60 km$^2$ Rio Behar catchment (67.047°N, -49.033°W) with flow <40 m$^3$ s$^{-1}$ in the main river stem leading to the terminal moulin, (Smith et al., 2017) report a lag time of 5.5 hours from Acoustic Doppler Current Profiler (ADCP) measurements from 20 to 23 July 2015, and Smith et al. (2021) report a lag time of 6 hours from repeated ADCP measurements on the same river segment from 6 to 13 July 2016. Our modelled inputs for moulins M1 and M3 (<40 m$^3$ s$^{-1}$ in 2011, 2015, and 2016) have similar input rates to the Rio Behar measurements and show a similar lag time (~4--7 hours). Lag times for M1 and M3 are higher in 2012 (~5--7 hours) when the absolute magnitude of moulin inputs is also much larger (up to >60 m$^3$ s$^{-1}$).

(ii) Section 4.2 is missing a description of differences in the timing of meltwater delivery to moulins with catchments that either have or do not have supraglacial lakes. What were the differences that the model produced?
Following the **General Changes**, this section and Fig. 3 will be updated to include an assessment of the controls on moulin discharge for the small catchments without supraglacial lakes. The primary findings are that the three small catchments in our domain without lakes have shorter lag times and larger diurnal amplitudes than the four large catchments with supraglacial lakes.

We have refocused the Results and Discussions related to the controls of lakes on moulin inputs to a qualitative comparison of the statistical relationships (Table 2, Section 4.3).

 (iii) There is no discussion of where the lake depth measurements are coming from, no discussion that justifies or explains why this model allows lake level to rise meters above the maximum lake surface.
We will make it clear that the lake depth measurements are derived from the model outputs. The "overfilling" of lakes is due to water flooding the lake drainage pathway, particularly for L1 which lies in a deep trough with low slope (L144-145) (e.g., Chudley et al., 2019). We now discuss the benefits that performing future in situ measurement of lake depth and outlet drainage path incision would have for model validation. Unfortunately these data are not currently available for this study site.

We advocate for in situ measurements of lake water level along with streamflow measurements in Lines 246 to 249:

However, given the currently available observational data, it remains difficult to calibrate the model and determine the most relevant mechanisms by which supraglacial drainage evolves. A full calibration dataset would ideally contain streamflow observations spanning multiple orders of discharge magnitude, measurements of lake water level, and the evolution of channel cross sections, for a significant portion of the melt season.

4. Lack of comparison with data to show that the model is actually physically representative of ice sheet evolution. A description of how the model was validated would add credibility to the results presented in this manuscript. At present it is unclear if these results are physically meaningful.

Hill & Dow (2021) present an indirect comparison to a satellite-derived drainage system map (Yang & Smith, 2016). The authors are not presently aware of a dataset(s) containing stream discharge measurements for the entire melt season covering the range of stream discharges presented here (<1 $m^3$ $s^{-1}$ to >50 $m^3$ $s^{-1}$). This makes it difficult to validate the model. The most appropriate dataset is that presented in Muthyala et al. (2022). However, this dataset would only allow validation of the model for small streams, which have distinct behaviour from the large rivers that dominate our study site. Smith et al. (2021), for example, provide 168 hours of measurements for a larger river (5.75 to 37.61 $m^3$ $s^{-1}$). However, this dataset is not appropriate to constrain seasonal changes in the drainage system, which is the primary intention of the SaDS model. We hope that our work demonstrating the importance of seasonal surface drainage evolution will encourage collection of long-term data sets and we point this out in the manuscript in section 4.4.

See above for the discussion of the need for additional validation data (Lines 246 to 249).

5. Repeated use of imprecise language, some examples below:

L11 controls on mass loss are vague and glossed over, this important topic warrants specifics
We agree that this topic is important. However, since the current work does not directly address controls on mass loss, it would be misleading to focus on the controls on mass loss. See below (**General changes**) for our plan to restructure the Introduction to focus on the most relevant background material.

We have removed these statements and have refocused the Introduction on how supraglacial drainage controls subglacial hydrology, the understood processes that control supraglacial drainage, and contemporary supraglacial drainage models.

L16 "…can improve the efficiency", the concept of drainage system efficiency has not been introduced and the phrasing "improve efficiency" is not clear
We believe this terminology is sufficiently common in the context of subglacial hydrology to be presented without a complete definition.

L4 of the abstract "the model outputs predict important differences…"
L16 "meltwater to be evacuated at lower pressure", lower than what?

L27 "significantly affect"
L113 "noticeably later", is 4 days noticeable? State 4 days instead
L120 "significant seasonal changes"
L164 "uncorrelated" this needs to be shown, r values and p values needed
L147 "saturate", what does this mean? Consider rephrasing
These examples and the language throughout have been made more precise.

L151 "distributed and channelized systems" this terminology is used here to describe the supraglacial drainage system but it typically reserved for the subglacial drainage system. Further, the components of the supraglacial drainage system are not described in the introduction (I.e., flow through channels vs interfluvial flow). This needs to be rephrased.

We intentionally use this terminology, which the reviewer has correctly identified as borrowed from subglacial hydrology, since we intend to convey the exact same concepts. We will present key model equations in the Appendix and more fully describe the key model structure in the Introduction to give the readers the necessary context. To clarify for the readers, the difference between flow through channels and interfluvial flow has been explained in the Introduction.

We have explained the components of the drainage model in Lines 17 to 28,

> Water flow through the supraglacial drainage system is understood to occur through both distributed, or hillslope/interfluve, drainage (e.g. Pitcher and Smith, 2019), and through discrete supraglacial channels. The density of supraglacial channels (which we refer to as "drainage density") is a key control on the amplitude and timing of moulin inputs, since transport through channels is more efficient than through distributed drainage (Yang et al., 2018; Pitcher and Smith, 2019; Yang et al., 2022).

and their representation within SaDS in Lines 61 to 66:

> The model is posed on an unstructured triangular mesh, with hillslope flow across elements, channelized flow on edges, and moulins on nodes. [...]

**Line Comments**
23: extreme surface melt events and lake drainages would overwhelm an inefficient drainage system as well.
That is correct, these events with unusually high moulin input rates would result in a transient high water pressure response for any subglacial drainage configuration. As part of the **General Changes**, we are focusing the Introduction more specifically on the supraglacial drainage system, so this statement will be removed.

This statement has been removed

27: "damp and delay" is introduced here but never referred to again,
This sentence will be reworded into more plain language.

This statement has been reworded (Line 28):

> Supraglacial drainage characteristically acts to reduce the diurnal amplitude and delay
> the timing of moulin inputs relative to the diurnal cycle of surface melt.

32: lag time between what and what?, also include a citation to Yang 2018, or say "for example".
Lag time between local solar noon and peak discharge. We will explain this in the text.

See the above modification.

55: "triangular computational mesh" is not introduced and is therefore confusing. Consider
flipping the order so that your model description is before the data subsection.
We agree this is confusing. We will reorganize this section as suggested, explaining the model
before the input data.

We have reversed the ordering of the model description (Section 2.2) and data description
(Section 2.3).

65-68: RAMCO melt rate with a 3 hour resolution is downsampled to 20 seconds for the SaDS
model run. How does this affect your results? What is the temporal error associated with your
lag times? This needs a more thorough discussion
We have tested smoother interpolation schemes to downscale RACMO melt rate to the 20
second model timestep, and there is a minimal change in the timing of peak moulin inputs
(0–0.5 hours) and a ~1% change in peak moulin inputs compared to a linear interpolation
scheme. We therefore believe the temporal error associated with lag times is small compared
with the overall uncertainty arising from using climate model forcing and will explain this
comparison and associated uncertainty in the text.

We have added a discussion of these tests to Lines 78 to 83 and presented the results in Figure
B1.

> Based on tests applying RACMO melt with linear and piecewise cubic modified Akima
> interpolation (Akima, 1970) (implemented with the MATLAB option 'makima'), the
> discrepancy in the timing of peak moulin inputs is <0.5 hours and the discrepancy in the
> magnitude of peak moulin inputs is ~1% between linear and Akima interpolation (Fig.
> B1). Therefore, since the model outputs are only weakly sensitive to the interpolation
> scheme, and since we don't have information on how melt rate varies on timescales
> shorter than the RACMO timestep (3 hours), we use the simplest numerically feasible
> option and linearly interpolate melt rates in time.

73: Describe what a distributed water sheet is and how this relates to physical flow structures

On the bare-ice surface, the distributed water sheet refers to flow through the weathering crust. We will add this to the model description in the Introduction and in the Appendix where we will present the key model equations.

The model is now described in Appendix A.

75-76: This triangular mesh/model domain is not shown in Figure 1 as cited. A figure with the model domain needs to be included in either the manuscript or at minimum in the supplement. We agree that it was difficult to see the triangular mesh. We have added translucent lines for triangle edges on Figure 1b.

86: Define what you consider a melt event early on in this section, it would make it easier to understand the text and to distinguish from a diurnal melt cycle and a multi-day melt event. We will make this more precise by referring to specific dates representing archetypal events in the text.

We have removed the imprecise language throughout Sections 3 and 4, and refer to specific dates and values.

95-100: Change the order of your examples so they are chronological, also what moulin are you referring to? Moulin input values are given but no specific moulin is named. We will make this more precise. To make the rest of the Results and Discussion more precise, we will assign numeric labels to each catchment, so that moulins and lakes can be explicitly related, and so the text can refer to specific catchments. In this example, we were referring to the moulin downstream from lake L1.

We have labelled moulins and catchments M1 to M7, and lakes L1 to L4, and are specific when referring to a given moulin.

88-89: Here you state that moulin inputs broadly track surface melt with the volume dependent upon catchment size, moulin location and melt volume, citing Figure 2, however, this figure does not show this relationship. There is no quantified or demonstrated relationship between discharge and catchment geometry. The relationship was not made explicit, but the relationship is qualitatively demonstrated by comparing Fig. 1 & 2. We have defined the catchment boundaries and will add these to Figure 1. We have explicitly quantified the relationship between melt rate within each catchment and the corresponding moulin input, and will tabulate this within the text.

Table 1 summarizes catchment sizes and average melt rates, and Table 2 quantifies the relationship between moulin inputs and surface melt.

101: Having an amplitude of 100% doesn't make sense, and comparing melting to moulin inputs is similarly confusing. Discharge isn't going to zero because of recession flow. So these two things can't be compared directly.

Throughout the text we refer to the difference between the diurnal maximum and minimum moulin inputs as the diurnal amplitude, and we will add this definition to the text. With this definition, an amplitude of 100% means that the difference between diurnal maximum and minimum moulin inputs is equal to the diurnal mean moulin input. In other words, no surface inputs are provided to moulins overnight when the amplitude is 100%.

Comparing surface melt (i.e., the system's forcing) to moulin inputs (i.e., the system's response) illustrates how strongly reduced the diurnal peak moulin inputs are. We do not claim that we expect moulin inputs to reduce to zero overnight in general. This comparison is especially useful considering that it is an accepted approach to instantaneously route all melt generated within a catchment into the moulin for subglacial hydrology modelling.

146: Cascading lake drainage comes out of nowhere
We agree that this statement is not properly supported or explained and will remove it.

This statement has been removed

147-150: Lake water level does not affect the timing of hydrofracture. Consider removing this paragraph as it is incorrect.
We are not aware of evidence that the timing of rapid lake drainage is entirely independent of lake water level. For example, Poinar & Andrews (2021) find that variance in the date of fast lake drainage is not fully explained by strain rate variations. A nonzero proportion of the variance in the date of lake drainage is explained by the date of melt onset and cumulative surface melt, which together approximately determine the volume of water stored within lakes. We will therefore make this statement more precise and explain rapid fluctuations in lake drainage as a possible confounding effect, rather than a primary control, on rapid lake drainage.

To be clear that lake water level is not a primary control on lake hydrofracture, but potentially contributes as a secondary variable, this statement now reads (Line 259)

> These rapid fluctuations in lake level may partly explain the residual variance in the timing of rapid lake drainage not explained by strain rates (Poinar and Andrews., 2021).

155-161: Melting of the channel's base and walls by the flowing water is not discussed in this section. This process is particularly important for the channels draining supraglacial lakes which can incise several meters into the ice.
This mechanism is not discussed here since rapid lake drainage through incision of the outlet channel is not observed in our model outputs, despite the theoretical ability of the model to permit this behaviour. In part, this could be due to the necessary assumption of a fixed width-to-depth ratio for supraglacial channels. We are adding a discussion of physical mechanisms expected to control supraglacial drainage to the introduction, which will include this mechanism. We will point out the possibility of this rapid lake drainage mechanism when we compare SaDS to other drainage models, since the ability to represent this mechanism is a distinguishing feature of SaDS.

Section 4.3 discusses the spectrum of lake behaviours observed by Lakes L1 to L4 and possible drainage mechanisms, including rapid drainage by hydrofracture for L4 and drainage by channel incision for lakes L1 and L2.

169: correlation < 1? This doesn't make sense
What we intended to say is that not all the variance in moulin discharge is explained by lake depth. We have directly quantified the proportion of variance of moulin discharge explained by lake level to make this statement clear.

Lines 126-136 explain our approach to quantifying the correlation and significance between variables, and Table 2 presents the results.

178: The other models discussed in this section need to be introduced in the paper's introduction, at present they come out of nowhere.
As part of the **General Changes**, these models will be presented in the Introduction in order to motivate the comparison.

These models are introduced in the Introduction in Lines 28 to 32.

213: "seasonal decreasing trend in moulin diurnal amplitude", do you mean moulin inputs? This is a very important distinction that needs to be made as you do not discuss moulin water level at all.
Yes, we mean the diurnal amplitude of moulin inputs. We are not making any statement about the absolute water level within moulins. We have reworded this statement accordingly.

This statement has been removed. We have used "the relative diurnal amplitude of surface inputs to moulins" throughout (e.g., Line 101) to make it clear what we are describing.

218-219: it is not clear that the moulin inputs presented here are realistic.
This is strictly true since we have not directly validated the presented hydrographs. To address this concern, we explain what type of data would be required to validate moulin hydrographs generated with SaDS in section 4.4, where we also qualitatively compare the two small, lake-free sub-catchments within our model domain to the Muthyala et al. (2022) dataset. Despite the lack of direct validation of our moulin hydrographs, the hydrographs contain specific features (lag time, diurnal amplitude) that are in line with observations, indicating that our moulin inputs are more realistic than the common practice to instantaneously route all meltwater generated with a catchment through the terminal moulin.

To be more careful, we focus on the level of detail in the processes resolved by different models, instead of claiming outputs to be realistic (Lines 305 to 309).

> A process-resolving model such as SaDS is expected to be advantageous when variations in the amplitude and timing of moulin inputs on sub-daily timescales will

substantially change the outcomes of the prospective modelling study. When only a broad representation of moulin inputs is required, simpler routing models are expected to be appropriate and come with a substantially reduced computational cost.

Figure 1, Outline individual catchments draining each moulin. At present I cannot tell which moulin is draining which lake, or visually compare catchment sizes. There is also not a triangular mesh as cited in the text. Also include a legend. Further, there is no explanation for why only three of the lakes are labeled when in Figure 1a I can see several other lakes within the catchment's boundaries. What is the pink moulin draining? I do not see any stream flow lines leading to that moulin.

The SaDS model does not explicitly differentiate between individual catchments in the same way as, e.g., the SRLF model (Banwell et al., 2012). We have defined catchments based on the modelled hydraulic potential, and will include the catchment boundaries on Figure 1.

We will add the computational mesh to Fig. 1, and we will make it clear that only the four largest modelled lakes are labelled. The identified moulin receives only very minimal flow (Fig. 2). This moulin is included since it is identified by Yang & Smith (2016), although it is possible it has been misclassified in that dataset.

Catchments draining through each moulin have been defined and shown in white in Figure 1b and summarized in Table 1.

Figure 2, Add a legend indicating what colors mean. The dashed/dotted line for melt rate is hard to see. Consider either making it solid black or another color. Right now it looks gray. (e.f.) Rename axis to state that the lag time is between solar noon and peak moulin discharge. The colors are very hard to see in general, particularly for (a) and (b), I can hardly tell that there are discharges plotted for pink and yellow moulins close to zero. It is also hard to tell the difference between the teal and green colors. Include which moulins are draining lakes and which are not in the legend as well. The color choice for lakes are also confusing, do the colors correspond with the moulins that drain them? If so, state that.

We have developed a consistent color scheme between lakes and their outlet moulins, and we will add the legend to Figure 2. We have also increased the linewidth for the melt rate curve to make it more visible. It is difficult to plot a two-order-of-magnitude difference in discharges on the same figure (a, b), but we have scaled the amplitude such that the amplitude trends for all seven moulins can be seen for (c, d).

Figure 3, This figure is not introduced until the discussion.

As part of the general restructuring, this figure will be removed in favour of a table quantifying the relationship between surface melt, moulin inputs, the diurnal amplitude of inputs to moulins, the time lag of moulin inputs, and the water level within supraglacial lakes.

This figure has been replaced by Table 2, summarizing the statistical relationship between key variables.

**Editorial Remarks**

L14: Change "This" to "The"

L88: the surface -> the ice surface

L92: "limited short-term (several day) spikes" -> several day spikes

L97: moulin inputs and moulin discharge are used interchangeably, chose one and stick with it

L97: which moulin?

We will make these changes.

These changes have been completed.

**General changes**

To address the general and specific comments above, and those of Reviewer 2, we propose to revise the manuscript as follows.

**Introduction**

- Reduce the discussion of the impact of meltwater supply variability on sliding velocity, since this is not directly supported by our work.
- Introduce and describe the models that are currently introduced in Section 4.4.
- The primary goals of the communication will be to:
  - Investigate the detailed modelled drainage behaviour for a catchment containing small catchments without lakes and large catchments with lakes.
  - Quantify the relationship between surface melt and the magnitude, amplitude, and timing of moulin inputs, and between supraglacial lakes and their outlet moulins.
  - Compare SaDS to a range of comparable models from the literature.
  - Define the situations where a process-resolving model may be advantageous and where such a model may not be practical.

**Data and Methods**

- Switch the order of Section 2.2 (Data) and 2.3 (Model) to describe the model before explaining the data used to drive the model
  - Explain key model mechanisms (e.g., balance between heat dissipation along channels walls and ablation of adjacent ice surface) in more detail in the model description section
  - Summarize the key model equations in an Appendix
- Explain that we model four years with varying melt intensity and melt season durations to capture how drainage behaviour varies with different realizations of melt forcing
- Assign consistent labels between moulins and lakes, and explicitly include these labels in the text when discussing particular catchments, moulins, or lakes

**Results**

- Use catchment labels to make discussion of particular features more precise

- Quantify relationships that are currently described qualitatively (e.g., computing the proportion of variance of moulin input, lag time, diurnal amplitude, and lag time that is explained by surface melt rate)
- Compare quantities of interest (moulin input, amplitude, lag time) for catchments with and without lakes and quantify the extent to which changes in lake level determine inputs to downstream moulins, and what the controls are for catchments without lakes

**Discussion and Conclusions**

The Discussion and Conclusions will center around a few questions supported by our Results:

1. What seasonal trends are observed in the modelled supraglacial drainage system? How do these vary with melt forcing?
2. How do supraglacial lakes impact modelled moulin inputs?
3. How do these modelled inputs compare to observations by Muthyala et al. (2022), in particular for our smaller catchments that do not have a supraglacial lake?
4. How does the behaviour we see across years with varying melt forcing compare to what would be predicted with other models? When might it be important to use an expensive process-resolving model, and when might it be appropriate to use a simpler and less computationally expensive model?

**Author Response for "Brief communication: The Impact of Interannual Melt Supply Variability on Greenland Ice Sheet Moulin Inputs"**

Tim Hill & Christine F. Dow

| Reviewer Comment | Author Response |
| --- | --- |

**Reviewer 2**

**General Comments**
This is an interesting modeling paper in which the authors simulate and compare the meltwater flow routing on a specific drainage area on the Greenland Ice Sheet for four different melt seasons. This paper specifically highlights the role of supraglacial lakes in the surface water routing delays on the bare ice ablation area of the Greenland Ice Sheet. This study is particularly relevant because of the lack of discharge data measured on the ice sheet, and because of the modeling challenges caused by the constant evolution of the landscape. Therefore, it is exciting to see a modeling study attempting to improve our understanding of what controls the amplitude and timing of peak discharge in moulins.

We appreciate the reviewer's interest in our manuscript. The comments are addressed individually below, followed by a high level plan to address both reviewer's comments (**General Changes**).

We have implemented the plan described in **General Changes** to address the comments by both reviewers.

However, there are several points I believe should be addressed:

There is a mismatch between the title, abstract, and introduction sections and the result and discussion sections.
We agree that the focus of the manuscript was not entirely consistent between sections. See the proposed **General Changes** related to this comment in addition to our specific responses below.

- The abstract suggests that the focus of the paper is on comparing high and low melt years, but there is no section about this comparison in the discussion. A comparison appears in the results, but no clear difference is demonstrated. How are high and low melt years defined? Is the difference between high and low melt years significant? If yes, what implication does this have for moulin and subglacial drainage dynamics? What are the other findings in this paper?
  We did not find clear differences in the amplitude and timing of moulin inputs for all sub-catchments between years with different melt forcing. Instead, variations in melt intensity tend to lead to abrupt shifts in the magnitude and timing of moulin inputs. The

same pattern was found observationally by Muthyala et al. (2022). Since we have not found a clear difference related to overall melt volume, we will remove the motivation in terms of low and high melt years, and instead focus on using four years with different melt forcing to explore a more complete range of modelled moulin input behaviours than would be possible with just one year.

We will rewrite the abstract to motivate the major aspects of our work:
1. To investigate the possible (modelled) behaviour of moulin inputs, in terms of amplitude and timing, in response to different patterns of melt forcing
2. Compare SaDS to other supraglacial hydrology models
3. Suggest a framework for selecting an appropriate supraglacial drainage model that depends on the level of detail and timescale of prospective modelling studies.

- The introduction suggests that the paper will compare the routing over different internally drained catchments during two pairs of successive melt seasons However, only two more extreme cases (one from each pair) are presented in the main text, and the analysis focuses only on the moulin drainage basins containing lakes.
Following from the comment above, the introduction will be rewritten to motivate the three listed themes to more fairly describe the goals of the remainder of the paper. We will also briefly describe the models we will later compare SaDS to, and will describe the key physical mechanisms modelled by SaDS. More generally, we have extended the analysis to include the moulin basins without lakes.
- The discussion section mainly focuses on the influence of supraglacial lakes over the damping, lag, and discharge into moulins, and not as the title suggests, how melt supply variability influences the moulin inputs.
We will integrate these sections of the paper by adjusting the abstract and introduction as described above, and by explicitly focusing the discussion and conclusions on the questions posed in the introduction.

These changes have been completed

The description of the results does not always differentiate between the main field site and the moulin drainage areas. For example, Figure S5 represents the entire area, instead of the individual catchments. In addition, the field site contains three moulins with a catchment with no lake and four moulins with a catchment draining a lake, and nearly only the moulins with the lake are described and discussed. An analysis of the moulin catchments without lakes would enable a comparison with the dynamics observed in the field data from Muthyala et al. (2022).

We have developed a consistent naming and colouring scheme between moulins and the lakes which they drain, and have added the corresponding catchments to Figure 1. We will update the text to refer to specific sub-catchments when appropriate.

We have analyzed two of the small, lake-free catchments to provide a more robust comparison to the field data from Muthyala et al. (2022). This comparison will form one of the main discussion points.

Section 4.4 compares two of the small, lake-free catchments to the observations of Muthyala et al. (2022). The main findings are that the diurnal amplitudes are similar, however a direct comparison is not possible given data availability.

The paper needs a clearer statement of the main findings and how they relate to our current understanding of supraglacial drainage systems and the simplifications made in models. Try to suggest appropriate usage for the model, and how it can be applied to a different field site. Are there any findings in this paper that could improve the simplified routing models that you mentioned in Section 4.4?

See the proposed **General Changes**. There, we suggest a few specific questions we will clearly answer that will make the main findings clear. Specifically, the main findings will be:
- The influence of model-induced variability in moulin inputs (i.e., changes in flow velocity, lake storage, and the extent of the supraglacial channel network that impact the amplitude and timing of moulin inputs) is the strongest in years with lower melt rates, and weakest in 2012.
- The four large catchments with supraglacial lakes have consistently lower diurnal amplitude in moulin inputs and peak moulin inputs occur later than for the three small catchments without supraglacial lakes. However, we can not fully disentangle the effects of catchment size from supraglacial lakes.
- The observational record presented by Muthyala et al. (2022) provides an important template for future field studies which, for the purpose of validating the model, should investigate streams with a range of discharge magnitudes within catchments that do and do not contain lakes.
- The appropriate supraglacial drainage model should be chosen based on a proposed modelling studies sensitivity to the amplitude and timing of moulin inputs, and the timescale in which variability in moulin inputs may impact the modelling results.

The Introduction and Conclusion now clearly state the goals and findings of the paper, as described above.

I suggest reformulating the title and modifying Sections 1-3 to match the current discussion findings, and redefining the scope of the results and how they fit in the current modeling landscape of the ice sheet. Alternatively, the results and discussion could extend to all the internally drained catchments simulated in this study and compare the consecutive melt seasons with each other. After reorganization and revisions, I believe this work would be of great interest to many readers, including myself.

We thank the reviewer for their helpful suggestions. See the proposed **General Changes** for our plan to refine the scope of the current work to reach the conclusions listed above.

**Specific comments**

The catchment is described as entirely within the bare ice zone (L47-48). However, it is likely that the drainage basin is covered by snow at the beginning of the melt season. What would be the potential influence of snow on the drainage basins at the beginning of the melt season until it is fully melted away? This is not currently discussed in the paper.

This is a good point, and we agree that at least part of the study site is snow-covered at the beginning of the melt season.
- Higher snow albedo than ice albedo would reduce melt rate for snow-covered elements
- Lower percolation velocities at base of snowpack compared to across bare ice (e.g., Arnold et al., 1998) would further reduce the diurnal amplitude & increase the lag time of streamflow
- Retention within the snowpack would delay meltwater transport until the snowpack melts. The previously retained runoff would be released over the duration of the final snowmelt event, resulting in a strong melt pulse in the early melt season.

We will add a discussion of these impacts to the revised manuscript and note that future development of the model could include processes emulating flow through a snowpack.

We have discussed these implications in Section 2.4:

In all years, SaDS is initialized with zero water depth across the supraglacial drainage system and with a bare-ice surface. This initialization neglects the impact of early season snow cover and interannual water storage in supraglacial lakes (Law et al., 2020) on moulin inputs. While seasonal snow cover persists, surface meltwater should be retained within the snowpack, with lateral transport occurring within saturated layers within the snowpack (e.g., Colbeck, 1974), therefore delaying early season moulin inputs. A physics-based snowpack and firn model (e.g. Meyer and Hewitt, 2017) is an attractive candidate to combine with SaDS. Such a model would additionally allow SaDS to be applied across a larger elevation range and more robustly in the early melt season. However, since this type of snow model would further increase the computational cost and significantly complicate the input data requirements of SaDS, this coupling is not pursued here.

In general, in the text and the figures, it is not always clear when the analysis is over the total catchment and when it is focused on a specific internally drained catchment. For example, most figures display separate timeseries for each moulin catchment, while Figure S5 seems to represent the entire study area. In addition, In the result and discussion, it is often unclear which catchment is being referred to. For example in line 97, specific numbers are given, but it does not say which moulin catchment it refers to. It would be helpful if the authors provided a logical numbering for the moulin catchment, displayed a legend on the timeseries, and referenced those catchments in the text.

We have developed a consistent naming and colouring scheme between moulins and the lakes which they drain, such that, for example, lake L3 drains through moulin M3. Catchments 1–4 will contain lakes, and catchments 5–7 will be the three smaller lake-free catchments. We will use

this numbering to add a legend to the timeseries figures and to be more specific in the text which catchment(s) we are referring to.

See Figure 1 and Table 1 for the consistent naming scheme.

Section 4.2 is very interesting. However, it is missing an analysis of the controls on lags in catchments without lakes. The authors could use the simulation on the smaller lake-free internally drained catchments to compare the results with Muthyala et al. (2022) (L140-141).

The reviewer raises a good point. The primary influence of supraglacial lakes appears to be to reduce the amplitude of moulin inputs and delay the timing of peak moulin inputs. Apart from these factors, the controls on moulin inputs appear similar for catchments with and without lakes. For example, we have quantified the extent to which surface melt can explain variance in moulin inputs, lag time, and moulin amplitude, and we do not find a difference in the relationship for catchments with and without lakes. We will add the corresponding content to the Results and create a new section in the Discussion to explain this more fully. We will also compare the modelled seasonal trends in amplitude and lag time to the trends reported by Muthyala et al. (2022).

Section 4.3 discusses the impact of supraglacial lakes on moulin characteristics, and what other processes are important for catchments without lakes. Section 4.4 compares two of the small lake-free catchments to Muthyala et al. (2022).

The introduction is missing a description of the physical mechanisms that are expected to control the evolution of the system. What is the known or expected behavior of the supraglacial routing system? How does the drainage system evolve? How channelized is it? Does the drainage system get more efficient? What controls the expansion/reduction of the channels/lakes? How do the model results compare to your expectations based on the physical mechanisms?

We will add a brief description of the physical mechanisms we attempt to capture with the SaDS model within length limitations of the current format. See also the proposed **General Changes** to refocus the Introduction to align more closely with the Results and Discussion sections. Some of the mechanisms we expect to control the drainage system are:
- Downward incision of supraglacial streams by flow-related potential energy dissipation and by shortwave radiation penetration (particularly for shallow streams). SaDS currently only represents potential energy dissipation.
- Ablation of the ice surface (in the modelling context, this specifically means ablation of 'elements') according to the local energy balance.
  - We expect that the balance between incision and ablation controls the extent of the channelized drainage system
- Rapid supraglacial lake drainage (e.g., Das et al., 2008)
- Slow overspilling of supraglacial lakes as the water level reaches the outlet level (e.g., Chudley et al., 2019)

- Snow plugs in streams and moulins (e.g., St. Germain & Moorman, 2019)
- Development of asymmetric "canyons" associated with deeply incised supraglacial streams (St. Germain & Moorman, 2019).

Lines 18-32 discuss the relevant physical mechanisms and their representation in models as described above.

For specific questions:
- What is the known or expected behavior of the supraglacial routing system?
  At the basic level, the supraglacial drainage system acts to reduce peak moulin inputs and delay the peak timing. We have described this in the introduction.

  Lines 28-29 describe the basic behaviour:
  > Supraglacial drainage characteristically acts to reduce the diurnal amplitude and delay the timing of moulin inputs relative to the diurnal cycle of surface melt (e.g., Smith et al., 2017; Muthyala et al., 2022).

  And the remainder of Lines 18-32 discuss the relevant mechanisms.
- How does the drainage system evolve? How channelized is it? Does the drainage system get more efficient? Does the drainage system get more efficient?
  It is not yet fully determined how drainage efficiency evolves. Yang et al. (2018) parameterize drainage density as decreasing through the melt season (e.g., drainage efficiency decreases). Based on satellite imagery, Yang et al. (2022) suggest a linear relationship between drainage density and runoff, and suggest this relationship can be used throughout the melt season to estimate drainage density, but do not explain the physical mechanisms that permit changes in drainage density on short (~1 day) timescales. Based on previous work with the SaDS model (Hill & Dow, 2021), large channels (discharge $>>$ 1 $m^3$ $s^{-1}$) grow larger throughout the melt season, since melt by heat dissipation along the channel bed and walls is faster than ablation of the adjacent ice surface, while small channels (discharge $<<$ 1 $m^3$ $s^{-1}$) melt out as ablation of the adjacent ice surface is dominant. The net effect is that more flow is captured by the large channels, but drainage efficiency decreases away from large channels.

  The expected evolution of the drainage system is described in Lines 18-28 and Appendix A.
- What controls the expansion/reduction of the channels/lakes?
  The evolution of channels depends on the balance between melt along channel perimeters and ablation of the adjacent ice surface (e.g., St. Germain & Moorman, 2019). The volume of water stored in lakes depends on difference between the rate of inflow and outflow (which may be due to rapid lake drainage (e.g., Das et al., 2008), rapid incision of a supraglacial spillway (e.g., Kingslake et al., 2015), or gradual overtopping (e.g., Chudley et al., 2019)). SaDS includes these mechanisms except for rapid lake drainage, and they are described in the model description section with the key equations now included in an appendix

> The evolution of the channel network and the representation of supraglacial lakes in SaDS is described in Appendix A.

- How do the model results compare to your expectations based on the physical mechanisms?

  Given the variability in surface melt rates, our model results do not clearly display how moulin inputs change with seasonal drainage evolution. This is unexpected based on the work of Yang et al. (2018) and Yang et al. (2022), as well as synthetic modelling with SaDS (Hill & Dow, 2021), but is in line with in-situ streamflow observations (Muthyala et al, 2022). We will explore this in the discussion.

  > Section 4.1 explores how surface melt rate is related to moulin inputs, and that seasonal trends observed in synthetic modelling (Hill & Dow, 2021) and observations (Muthyala et al., 2022) are not clear in the present study.

**Line Comments**

L 15-19: Andrews et al. 2014 demonstrate that the water level in moulins did not drop as expected previously, and found that it was the connectivity of the unchannelized portion of the bed that controls the seasonal variability of ice flow.

We see that the wording here was not as precise as it should be. Since we do not directly address the subglacial water pressure response to our modelled moulin inputs, we will reduce this discussion to a simple statement that subglacial hydrology impacts basal motion with a few key references (e.g., Bartholomew et al., 2012; Sole et al., 2013; Andrews et al., 2014).

This discussion has been reduced to

> The supraglacial drainage system acts as a mediator in the relationship between surface melt and subglacial water pressure (e.g., Bartholomew et al., 2012; Sole et al., 2013; Andrews et al., 2014) since meltwater generated at the surface must first flow through supraglacial catchments before being transported through the depth of the ice via moulins. Since supraglacial drainage in part controls the form of inputs to moulins, it has the potential to significantly affect the relationship between surface melt and ice velocity (e.g., Banwell et al., 2016).

L 43: How are high, average, and low melt years determined? Is it related to the mean discharge or the intensity of the peaks?

We will explain our definition of "low", "average", and "high" melt years. We define these melt years based on the total seasonal melt volume within our study site.

Since we don't observe clear differences in the behaviour of moulin inputs related to seasonal melt volume, we have removed the definitions of "low", "average", and "high" melt years.

Figure 1: The drainage basin displayed is small. It would help to have the moulins numbered on the figure to match the timeseries on the other figures. Sometimes the black outline is not very visible.

Catchments, lakes, and moulins will be numbered, and a legend will be added to the timeseries figures to relate the timeseries curves to the lake and moulin positions. We are also adding the boundaries for the seven supraglacial sub-catchments to this figure.

This has been completed.

L 67-69. in the results and discussion, only 2012 and 2015 are displayed in the main text. In addition, the pairs 2011-2011 and 2015-2016, are not compared with each other in the text.

As part of addressing the **General Comments**, different melt seasons will be explained as being used to investigate the full spectrum of drainage system behaviour. Since 2012 and 2015 are the years with the highest and lowest total melt, we will justify our focus on these in the main text.

Figure 2 has been separated into four separate figures to display all four years of model outputs.

Figure 2: The surface melt is defined as black in the caption, but it appears grey and dotted to me. There is a lot of information displayed on each subfigure. Would it be possible to plot the melt rate in a separate subfigure at the top instead of on each subfigure? The addition of Figure S5 to this graph would be nice too.

We agree that there is a lot of information displayed on each subfigure. However, we believe it is important to plot the melt rate in each panel since it allows the timing of changes in surface melt and the displayed quantities (discharge, amplitude, lag, lake level) to be directly compared. We will update the caption to correctly describe the melt rate line style. We have shifted our focus away from seasonal changes in channel extent, since the consequences of channel evolution are not the primary feature observed in moulin inputs, so we do not believe it's relevant to include Figure S5 here.

We have separated Figure 2 into four separate figures, with each showing the quantity of interest and surface melt rate. Since we do not focus on seasonal changes in the channel network, Figure S5 has been moved to Appendix C.

L87: Only 2012 and 2015 are displayed in the main text, but 2011, and 2016 are mentioned.

We will add a reference to the appropriate supplementary figures. Only 2012 and 2015 are displayed since they represent the highest and lowest total discharge.

Figure 2 has been separated into four separate figures to display all four years of model outputs.

L88-89: Could you display the catchment properties for each moulin? (maybe a table with name, catchment size, catchment melt volume, and size of the lake if present). This would help

compare the variability between the different internally drained catchments of your study. Does the catchment size for each moulin vary from one year to the other, and throughout a melt year? The SaDS model does not explicitly differentiate between individual catchments in the same way as, for example, for SRLF model. We have been able to define catchments from the modelled hydraulic potential, and we do not see changes in catchment size between years or throughout a melt year. This may be because the elements are large enough that the difference in elevation between neighbouring elements is greater than the maximum water thickness, such that flow directions are primarily determined by surface topography.

We do not have space in the main text for a table with catchment details, but we will add this to the supplement, as we agree it would be useful.

Table 1 now summarizes the catchment name, size, melt rate, elevation, and lake properties (if present).

L89-90: Could you elaborate on "Multi-day increases in melt rate cause a 1–3 day lag in peak moulin input…"? Where specifically on the figure do you see this? We propose to focus the Results section more on the amplitude, lake level, lag time, and how this differs with catchment size and the presence of lakes to make the results more quantitative and specific than, for example, the identified sentence. In this case, we were referring to the highest discharge values observed in 2012, which occur ~2 days after the peak in surface melt rate.

We have made the Results more specific by including Table 2, summarizing the relationship between variables of interest for each catchment, and identifying the dates and catchments for identified features. For example, Lines 108–111.

L90-91: "… with adjustment in the extent and size of incised supraglacial channel...". Are you referring to Figure S5? Yes, we are indirectly referring to Figure S5, Figure 2g, h, and Figure S4.

We have removed this statement, and Figure S5 has been moved to Figure C1, and have been explicit when identifying the role of changes in the channel extent. For example, Lines 182–183:
>   The lack of continuous trends suggests that variations in surface melt rate are the primary control on moulin inputs compared to seasonal expansion and contraction of the supraglacial channel network (Fig. C1).

L 92-100: It is unclear which moulin basins are referred to. We will add reference to the specific catchments we are referring to (L1).

The Results have been rewritten, so this statement has been removed. We identify specific basins throughout the Results and Discussion.

L93: Figure 2a is mentioned but 2012 is not included in the sentence

We will remove the reference to Fig. 2a.

L95-97: "... frequently recurring intense melt events result in persistently high discharge into moulins with large differences between minimum discharge… ". Is the discharge really persistently high in 2012 compared to the other years? It seems to me that the discharge in 2012 drops when there is lower melt production, similar to the other years.
We agree that this sentence was not precise. We propose to focus instead on the specific instances of large changes in moulin inputs described in L97-100.

We have removed this statement.

L 101: Did you mean "... nearly 100%..." of the mean melt?
Our wording is unclear here. We mean that surface melt pauses overnight. This is in contrast to the behaviour of model inputs described in the following lines.

This paragraph has been removed.

L 103: Is the diurnal amplitude a peak-to-peak amplitude? Could you discuss how the percentage of the diurnal amplitude of the mean discharge compares to the same ratio calculated with field data? For example, in Muthyala et al., (2022) the peak-to-peak amplitude seems to be nearly two times larger than the mean discharge.
We defined diurnal amplitude as peak-to-peak and will add this to the text. To compare to Muthyala et al. (2022), we will scale the amplitude by the mean moulin input in Fig. 2 (c, d) and focus on the small, lake-free catchments.

Section 4.4 compares the diurnal amplitude as a percentage of the mean input for two of the small basins to Muthyala et al. (2022) (Line 224 to 227).

> For M6 & M7 (combined catchment area of 8.5 km$^2$), we have modelled lag times between 2--5 hours, compared to 1--3 hours from Muthyala et al. (2022). SaDS predicts a relative diurnal amplitude of 50--100%, i.e., flow nearly pauses overnight, in qualitative agreement with data collected by Muthyala et al. (2022) showing stream discharge minimums near zero except during periods of elevated positive overnight energy balance.

L110-111: How do you know when the lake is filled? Is there any model output that could be displayed to show the relative filling of lakes?
We infer that the lake has filled by comparing the lake level to the steady level at the end of the melt season and by the downstream impact on moulin inputs (Fig. 3). We will explain this in the text.

Line 116 explains that

> [...] the storage volume in the absence of surface melt can be approximated by the lake depth at the end of the melt season [...]

L114: When is the initial onset of positive temperature? There is no figure displaying the temperature timeseries.

This was an imprecise statement where we implicitly used temperature as a proxy for melt rate. We will make this precise by simply using melt (shown in Fig. 2) directly.

L 117: what causes the melt out of small channels? It is unclear to me why the small channels get created at the beginning of the melt season and then melt out. I suppose I would expect an increase in channelization over the melt season, and therefore, an increase in the diurnal amplitude of discharge (L124-127). Is this a pattern observed in field measurements?

See our discussion of the mechanisms controlling evolution of the drainage system. Small channels melt out by ablation of the surrounding ice surface. The initialization with many small channels is a coarse way to represent channelization in snow and the impact of rapid snowmelt.

These mechanisms are described in Lines 18-32 and Appendix A.

The longest observational timeseries is that of Muthyala et al. (2022). As we find in our model outputs, Muthyala et al. (2022) don't observe obvious seasonal trends in the diurnal amplitude (Fig. 5a) due to variations in melt rate.

We advocate for observations to definitively answer this question, since it is not yet fully clear with available datasets. Lines 247 to 249 suggest that a wide variety of observations would be needed to constrain the mechanisms, including

> contain streamflow observations spanning multiple orders of discharge magnitude, measurements of lake water level, and the evolution of channel cross sections, for a significant portion of the melt season.

L 117-120: Consider moving this part to the discussion.

Following the **General Changes**, we are no longer focusing on changes in the stream network extent.

L 124: the end of the sentence "... attributable to changes in the extent of the supraglacial channel network." seems to refer also to Figure S5. However, Figure S5 only displays the years 2012 and 2015.

Following the **General Changes**, we are no longer focusing on changes in the stream network extent.

L 131-133: What about the lake-free moulin catchment (yellow and green situated NE of the study area? What is the size of those catchments and how do they compare with Muthyala et al.? The diurnal amplitude is not very visible for that catchment in Figure 2, due to the scale of the y-axis.

We have scaled the y-axis of the diurnal amplitude panels of this figure by mean discharge, so that the amplitude of all moulins can be compared in a meaningful way, including compared to the results of Muthyala et al. (2022) in the case of the small, lake-free catchments. This scaling reveals that these small, lake-free catchments have consistently higher diurnal amplitude in moulin inputs than the other large catchments with lakes.

L140-141: How does the lag time compare with Muthyala et al. (2022) and other field measurements from other studies for the internally drained catchments without a lake simulated in the study? (yellow, green, and purple moulins). For those lake-free catchments, what controls the lag times?

These are excellent points, and we plan to address these and other questions in a revised Discussion section (see response to the final **General Comment**). From preliminary work, the lag times in the catchments without a lake are 2–4 hours, where Muthyala et al. (2022) report 1–3 hours. For a ~60 km$^2$ catchment with peak flow < 40 m$^3$ s$^{-1}$, Smith et al. (2017) report a lag time of 5.5 hours, and Smith et al. (2021) report a lag time of 6 hours. These compare well to our modelled lag time for moulins with inputs >40 m$^3$ s$^{-1}$ of 4–7 hours (Fig. S3). We will add these details to the main text.

Section 4.4 presents these comparisons, with additional comparisons to Smith et al. (2017) and Smith et al. (2021).

L147. Could you elaborate on the saturation at the outlet elevation?

What we mean here is that the lake water level can transiently exceed the elevation of the lake outlet when inflow exceeds outflow. This means that the maximum hydrostatic pressure acting on incipient fractures on the lake bed can exceed the pressure that would be computed by assuming the lake level is exactly at the outlet elevation. We will make this clear in the text.

Section 4.3 describes the behaviours observed in the modelled lakes, including transient storage and slow draining by overfilling.

Section 4.3: Based on your results, when a lake is present on a drainage basin, would it be possible to calculate the discharge at the moulin based only on the lake elevation change and the lake area? Would such a correlation improve stream discharge estimation from satellite imagery?

For basins with lakes, we have computed that the correlation between lake level and downstream moulin inputs is between $R^2$=0.22 and $R^2$=0.92 (p < 0.01). However, the exact relationship differs between each lake basin, so any such relationship would likely need to be calibrated with model outputs or in-situ streamflow measurements for each basin.

Section 4.4: This section's purpose in the discussion is unclear to me. The comparison with other models is interesting, however, the takeaway message is unclear. Should this section maybe be in the intro or the model description?

The intended purpose of this section was to provide justification for recommendations on when an expensive process-based model such as SaDS is important, and where a simpler and more

efficient model (e.g., SRLF; Banwell et al., 2012) may be appropriate. However, we did not make this purpose or these recommendations clear. As part of the overall restructuring, appropriate background material from this section will be moved to the Introduction and our recommendations and justification will be clearly laid out. We suggest that the appropriate supraglacial drainage model should be chosen based on (1) the sensitivity of the proposed modelling study's results to the amplitude and timing of moulin inputs, and (2) the allowable computational cost allocated to the supraglacial model.

Section 4.5 now presents clear directions on selecting the appropriate supraglacial drainage model for a specified modelling objective, which is supported by the discussion of available models.

L157-161: Is there field evidence for overflowing supraglacial streams? Is this a frequent situation in your model simulations?
Exceptionally clean and smooth ice surrounding supraglacial streams may suggest that streams have overflowed in the current melt season. For example, see (b) and (c) https://www.antarcticglaciers.org/wp-content/uploads/2021/07/figure1_HR-scaled.jpg (credit J. Gulley) and Figure 1 from Smith et al. (2015). This is a frequent situation in our model simulations. This may be due to SaDS neglecting the contribution of shortwave radiation to melt along stream beds, or from neglecting the enhanced melt on the portion of channel walls above the waterline. Both of these processes are difficult to resolve since they should properly be a function of channel geometry, which we are forced to model as fixed to make the model numerically tractable. These limitations are discussed in detail in Hill & Dow (2021).

L165: Is there any way to display on the figures when lakes are filled? Is this a dynamic process?
For Figure 3, we could plot a vertical line or shade the region above which the lake is 'full' according to its outlet elevation. Unfortunately, we will need to remove Figure 3 given the length limitations.

This discussion has been removed in favour of a quantitative analysis of the statistical relationship between quantities of interest (Table 2).

L 203-204: Only two years are presented in the figures in the main text.
We will explain our reasoning for including two years (2012, 2015) in the main text in Section 3.

Figure 2 has been separated into four separate figures, each showing all four years, and the text throughout Sections 3 and 4 discusses each year.

Figure S1: Melt doesn't follow the same color coding as the figure in the main text. Is there a moving average in this figure? The second part of the legend might belong to Figure S2 ("Light colors show instantaneous diurnal amplitude, and bold colors show the seven-day moving average."). To what corresponds to the black line?
Apologies for the inconsistency in the color coding and the inaccurate caption.

Figure S1: The solid black line shows the 24-hour moving-average melt rate. There is no moving average performed on the discharge data.

Figure S2-S3: Light colours show instantaneous quantities and bold colours show 7 day moving-average quantities.

These captions will be corrected.

Figure S2 : There is not much difference between 'light' and 'bold' in the figure.
The lineweight of the solid line will be increased to make the distinction more clear.

Figures S1, S2, S3, and S4 duplicate the information displayed in Figure 2.
We acknowledge that some of the information here is duplicated from Figure 2. However, we think it is useful to have all modelled years present in these figures to save the reader from needing to move between the supplement and main text to compare certain years.

Figures S1–S4 have replaced Figure 2 in the main text.

Figure S5: This Figure only contains the years 2012 and 2015. In addition, they are not present in the main text. This would be a good addition to Figure 2.
The other years (2011, 2016) will be added. Since we are shifting focus away from seasonal evolution of the channel network, we do not think it is relevant to add the content from Figure S5 to Figure 2.

Figure S5 has been moved to Appendix C.

Technical comments

L 159-161. Consider breaking up this sentence. "... with a large increase...", ...'and stored in supraglacial..."
We will restructure the sentence as suggested.

Throughout the paper, there's a mixed-use of kilometer and meter. Example, L 50-54.
We will translate all lake surface and catchment areas to $km^2$.

**General changes**
To address the general and specific comments above, and those of Reviewer 1, we propose to revise the manuscript as follows.

**Introduction**
- Reduce the discussion of the impact of meltwater supply variability on sliding velocity, since this is not directly supported by our work.

- Introduce and describe the models that are currently introduced in Section 4.4.
- The primary goals of the communication will be to:
  - Investigate the detailed modelled drainage behaviour for a catchment containing small catchments without lakes and large catchments with lakes.
  - Quantify the relationship between surface melt and the magnitude, amplitude, and timing of moulin inputs, and between supraglacial lakes and their outlet moulins.
  - Compare SaDS to a range of comparable models from the literature.
  - Define the situations where a process-resolving model may be advantageous and where such a model may not be practical.

**Data and Methods**
- Switch the order of Section 2.2 (Data) and 2.3 (Model) to describe the model before explaining the data used to drive the model
  - Explain key model mechanisms (e.g., balance between heat dissipation along channels walls and ablation of adjacent ice surface) in more detail in the model description section
  - Summarize the key model equations in an Appendix
- Explain that we model four years with varying melt intensity and melt season durations to capture how drainage behaviour varies with different realizations of melt forcing
- Assign consistent labels between moulins and lakes, and explicitly include these labels in the text when discussing particular catchments, moulins, or lakes

**Results**
- Use catchment labels to make discussion of particular features more precise
- Quantify relationships that are currently described qualitatively (e.g., computing the proportion of variance of moulin input, lag time, diurnal amplitude, and lag time that is explained by surface melt rate)
- Compare quantities of interest (moulin input, amplitude, lag time) for catchments with and without lakes and quantify the extent to which changes in lake level determine inputs to downstream moulins

**Discussion and Conclusions**
The Discussion and Conclusions will center around a few questions supported by our Results:
5. What seasonal trends are observed in the modelled supraglacial drainage system? How do these vary with melt forcing?
6. How do supraglacial lakes impact modelled moulin inputs?
7. How do these modelled inputs compare to observations by Muthyala et al. (2022), in particular for our smaller catchments that do not have a supraglacial lake?
8. How does the behaviour we see across years with varying melt forcing compare to what would be predicted with other models? When might it be important to use an expensive process-resolving model, and when might it be appropriate to use a simpler and less computationally expensive model?

**References**

Andrews, L. C., Catania, G. A., Hoffman, M. J., Gulley, J. D., Lüthi, M. P., Ryser, C., ... & Neumann, T. A. (2014). Direct observations of evolving subglacial drainage beneath the Greenland Ice Sheet. *Nature*, *514*(7520), 80-83.

Arnold, N., Richards, K., Willis, I., & Sharp, M. (1998). Initial results from a distributed, physically based model of glacier hydrology. *Hydrological Processes*, *12*(2), 191-219.

Banwell, A. F., Arnold, N. S., Willis, I. C., Tedesco, M., & Ahlstrøm, A. P. (2012). Modeling supraglacial water routing and lake filling on the Greenland Ice Sheet. *Journal of Geophysical Research: Earth Surface*, *117*(F4).

Bartholomew, I., Nienow, P., Sole, A., Mair, D., Cowton, T., & King, M. A. (2012). Short-term variability in Greenland Ice Sheet motion forced by time-varying meltwater drainage: Implications for the relationship between subglacial drainage system behavior and ice velocity. *Journal of Geophysical Research: Earth Surface*, *117*(F3).

Chudley, T. R., Christoffersen, P., Doyle, S. H., Bougamont, M., Schoonman, C. M., Hubbard, B., & James, M. R. (2019). Supraglacial lake drainage at a fast-flowing Greenlandic outlet glacier. Proceedings of the National Academy of Sciences, 116(51), 25468-25477.

Das, S. B., Joughin, I., Behn, M. D., Howat, I. M., King, M. A., Lizarralde, D., & Bhatia, M. P. (2008). Fracture propagation to the base of the Greenland Ice Sheet during supraglacial lake drainage. *Science*, *320*(5877), 778-781.

Hill, T., & Dow, C. F. (2021). Modeling the dynamics of supraglacial rivers and distributed meltwater flow with the Subaerial Drainage System (SaDS) model. *Journal of Geophysical Research: Earth Surface*, *126*(12), e2021JF006309.

Kingslake, J., Ng, F., & Sole, A. (2015). Modelling channelized surface drainage of supraglacial lakes. *Journal of Glaciology*, *61*(225), 185-199.

Muthyala, R., Rennermalm, Å. K., Leidman, S. Z., Cooper, M. G., Cooley, S. W., Smith, L. C., & Van As, D. (2022). Supraglacial streamflow and meteorological drivers from southwest Greenland. *The Cryosphere*, *16*(6), 2245-2263.

Smith, L. C., Chu, V. W., Yang, K., Gleason, C. J., Pitcher, L. H., Rennermalm, A. K., ... & Balog, J. (2015). Efficient meltwater drainage through supraglacial streams and rivers on the southwest Greenland ice sheet. *Proceedings of the National Academy of Sciences*, *112*(4), 1001-1006.

Smith, L. C., Yang, K., Pitcher, L. H., Overstreet, B. T., Chu, V. W., Rennermalm, Å. K., ... & Behar, A. E. (2017). Direct measurements of meltwater runoff on the Greenland ice sheet surface. *Proceedings of the National Academy of Sciences*, *114*(50), E10622-E10631.

Smith, L. C., Andrews, L. C., Pitcher, L. H., Overstreet, B. T., Rennermalm, Å. K., Cooper, M. G., ... & Simpson, C. E. (2021). Supraglacial river forcing of subglacial water storage and diurnal ice sheet motion. *Geophysical Research Letters*, *48*(7), e2020GL091418.

Sole, A., Nienow, P., Bartholomew, I., Mair, D., Cowton, T., Tedstone, A., & King, M. A. (2013). Winter motion mediates dynamic response of the Greenland Ice Sheet to warmer summers. *Geophysical Research Letters*, *40*(15), 3940-3944.

St Germain, S. L., & Moorman, B. J. (2019). Long-term observations of supraglacial streams on an Arctic glacier. *Journal of Glaciology*, *65*(254), 900-911.

Yang, K., & Smith, L. C. (2016). Internally drained catchments dominate supraglacial hydrology of the southwest Greenland Ice Sheet. *Journal of Geophysical Research: Earth Surface*, *121*(10), 1891-1910.

Yang, K., Smith, L. C., Karlstrom, L., Cooper, M. G., Tedesco, M., van As, D., ... & Li, M. (2018). A new surface meltwater routing model for use on the Greenland Ice Sheet surface. *The Cryosphere*, *12*(12), 3791-3811.

Yang, K., Smith, L. C., Andrews, L. C., Fettweis, X., & Li, M. (2022). Supraglacial drainage efficiency of the Greenland Ice Sheet estimated from remote sensing and climate models. *Journal of Geophysical Research: Earth Surface*, *127*(2), e2021JF006269.

---

## Author Response (AR2)

**Author Response for "The impact of surface melt rate and catchment characteristics on Greenland Ice Sheet moulin inputs"**

Tim Hill & Christine F. Dow

| Reviewer Comment | Author Response | Manuscript changes |
|---|---|---|

**Reviewer 1**

**General Comments**

In the paper entered "the impair of surface melt rate and catchment characteristics on greenland ice sheet moulin inputs" the authors apply the SaDS surface meltwater routing model to a group of catchments located in west Greenland. The authors compare low and high intensity melt seasons to determine the relative importance of surface melting on meltwater inputs to moulins and the impact of supraglacial drainage system evolution. The authors find that supraglacial drainage system develop has a more pronounced influence on meltwater delivery to moulins in years with lower melt rates, while going on to provide recommendations for when to apply the computationally expensive SaDS model over other less expensive models. The authors have addressed many of the concerns raised in the last round of review and as a result the manuscript is much improved. I have included a few major concerns that should be addressed before publication, however once addressed I think he manuscript will make a significant contribution to the literature. A robust model such as the SaDS model will be increasingly significant as our in situ glacial hydrology observational record continues to expand.

We appreciate the reviewer's detailed comments and have responded individually below.

We would also like to highlight a change to the statistical analysis in Table 2. While investigating the reviewer's comments to Line 126-136 and Table 2, we found a mistake in the calculation of $R^2$ and p values. This has been corrected, all other code has been double-checked, and Section 4.1 has been updated accordingly.

**Major Comments**

In this paper lag times are presented as between solar noon (~15:22) and peak moulin inputs, rather than the lag time between peak melting and peak moulin inputs. Using the later definition would be inline with previous studies (). As is written now, it is unclear whether the timing of solar noon is allowed to change (presumably due to the approximate time stated in the manuscript), nor is it clear if there is a lag between the timing of peak melting and solar noon. This may become problematic for example if peak melting during transient melt events did not coincided with local solar noon (ref to the results presented on L113-115). If so, the longer lag

time in moulin input would be an artifact of the timing of peak melt rather than caused by the supraglacial drainage system.

We have compared the timing of peak moulin inputs to solar noon due to the limited temporal resolution (3 hours) of the RACMO data used to drive SaDS. Smith et al. (2021; Fig. 3) and Mejia et al. (2022; Fig. 2) suggest that the range in timing of peak melt is less than three hours (the resolution of the RACMO data), so we can not resolve these slight variations. Fortunately, Mejia et al. (2022; Fig. 2) suggests that comparing peak moulin inputs to local time vs. peak melt does not make a significant difference in the interpretation of the timing of moulin inputs. Furthermore, our approach is not unusual, for example Muthyala et al. (2022) compare the timing of peak stream discharge to local solar noon. We have acknowledged this limitation in the revised manuscript when comparing our modelled lag times to the time between peak melt and peak stream discharge from Smith et al. (2017, 2021) (Line 267-272):

However, these differences in lag time should be interpreted with caution since Smith et al. (2017) and Smith et al. (2021) report lag times relative to peak melt rather than solar noon. This difference could be important, for example, if local weather conditions modulate the timing of peak melt relative to solar noon (e.g., Smith et al., 2021). On the other hand, the difference in timing between solar noon and peak melt reported by Mejia et al. (2022) for 0.2 $km^2$ and 16.7 $km^2$ catchments is less than three hours, so we would not be able to resolve these differences with our three-hour resolution surface melt forcing data.

Statistical Analysis
A few things regarding the statistical analysis presented on lines 126-136 and in Table 2 are unclear. First, the $R^2$ typically represents the coefficient of determination whereas $r^2$ would be the square of the Pearson correlation coefficient. I assume the text is referring to the former as it is stated that $R^2$ is equal to the proportion of variance explained by the independent variable. I know this is very in nit-picking but it is important to be precise here. The Pearson correlation coefficient (r) should also be included in this analysis.

The reviewer is correct to point out the general distinction between the coefficient of determination ($R^2$) and the squared correlation coefficient ($r^2$). However, for the comparison as described here, we have computed these two quantities independently and verified they are identical. For clarity, we explicitly refer to the coefficient of determination ($R^2$) throughout the text. For example, Line 133 now reads

The extent to which surface melt rate controls these features can be quantified by comparing the coefficient of determination, $R^2$, between melt rate and each of the moulin input rate, diurnal amplitude, lag time, and lake water level (the coefficient of determination is equal to the square of the Pearson correlation coefficient, $r^2$, for linear regression).

In the same regards, I am confused by the stated maximum and minimum values for $R^2$ as there should be a single value given for each of the correlations.

We have stated that the comparisons are carried out for each of the seven sub-basins, meaning that we obtain seven $R^2$ and p values. We have chosen to report the minimum and maximum of these seven $R^2$ values in Table 2 to measure how these relationships vary by basin. The caption for Table 2 has been expanded to explain these values.

Table 2. Coefficient of determination ($R^2$) and p-values for the null hypothesis that there is no relationship between the specified variables. Coefficients $R^2$ and p-values are computed independently for each of the seven sub-catchments and for each year. The tabulated min and max $R^2$ values represent the minimum and maximum R2 values taken across the seven catchments for a given year, and the p values represent the maximum value across the seven catchments. Coefficients R2 and p-values are computed for model outputs at native 2-hour resolution and binned into 24-hour increments

Table S1 does not give statistics or p-values as stated in L134-L136.

We believe this confusion was caused by an old version of the supplementary material being provided to the reviewer. Table S1, accessible as described in the code and data availability statement, provides the stated statistics.

Additionally, even though p-values are small <10^-6 they should not be represented as 0 it is in violation of the definition of a p-value which is a probability.
The magnitude of all values are provided in the table.

And finally, there should be a figure added to the supplement showing these relationships as graphs are essential to correctly interpret regression analysis results.
We have added figures to Appendix B for each of the comparisons (Fig. B3–B7).

Regarding the interpretation of the statistical tests it is unclear how the p-values alone are being used to determine there are good correlations between variables while R^2 values range from 0.09—0.9, this issue here is not a low R^2 value but the variance between variables, years, and smoothing choices (e.g., diurnal vs daily). Moreover, I wonder if the lower R^2 values for the diurnal variables are a result of the lag time between variables. This is a common problem that is either solved by imposing a lag-time adjustment (e.g., Smith et al., 2021), or by instead analyzing forcing-response plots (e.g., Extended Data Figure 4 in Andrews et al., 2014). Due to the significant amount of text in the Results and incorporation within the Discussion (e.g., L165-170), I recommend explaining this analysis in more detail.

The additional scatter plots that we have now included (Fig. B3–B7) aid in the interpretation of the $R^2$ and p values. We agree that part of the reduced $R^2$ from two-hour resolution model outputs is due to the previously computed lag time. However, given the relatively little Discussion text devoted to the two-hour relationships and the additional context provided by the scatter plots, we believe it is interesting to evaluate the relationships as-is. The influence of lag time on these $R^2$ values is acknowledged on line 149:

The lower $R^2$ values obtained with two-hour model outputs may in part be due to the time lag between peak melt and peak moulin input rates.

In section 4.1 the manuscript states internal variability is most important on timescales shorter than one day, as evidenced by the statistical analysis. Is this conclusion supported by model results? Specifically, how are model parameters (e.g., channel water depth, incision depth, density, flow, etc.) changing on daily vs. sub daily timescales? In high melt vs low melt years (e.g., 2012 vs other years)? Figure C1 shows that there is diurnal variability in channel length, so how does this fit in? What is the breakdown between catchments (e.g., is this only important for large or small catchments? What controls this variability and how does this vary between years?

Channel flow metrics (water depth and discharge) change with similar characteristics as the moulin hydrographs, while sheet metrics (water depth, discharge) change with similar characteristics as the lake water level curves. Figure C1 shows the incision depth of supraglacial channels. This is similar for all catchments. While we appreciate the reviewers interest here, since this section is dedicated to interannual changes, we have removed this statement as it distracts from the intention of the section.

In the discussion comparing model outputs to other works from Rio Behar catchment lag times are compared to work by Smith et al., 2017. It is important to note here that the lag time reported in that paper are the time between peak melting and peak moulin inputs (this is different from how lag times are described in the present manuscript), and are accordingly not directly comparable.

See the response to the first major comment. This limitation has been acknowledged in Lines 267-272.

The discussion also describes the models limitations on refining the potential influence of supraglacial lakes on moulin inputs, is there a way to look at the outlet channels that drain the lakes and compare changes within those to other parts of the supraglacial drainage system to see if there are localized effects there on the draining lake? Alternatively, how do the lag times for catchments with lakes compare to a simple parameterization such as that used in Smith et al., 2017 (already cited within the manuscript)? I think understanding the influence of lakes on lags and meltwater inputs to moulins is very interesting and would be a significant contribution to the field of supraglacial hydrology. While I understand that lakes cannot be disentangled from your model domain, I wonder if comparison with a synthetic unit hydrograph could help parse out the lake's influence on lag times. (As stated previously, I would suggest redefining the lag determination used in the text to be consistent with other models (e.g., SUH/UH models).

We also believe there is more work to be done in evaluating the impact of lakes on moulin inputs, and we hope the current work and model can provide a starting point for such analysis. However, we struggle to constrain the relationship further with the currently available data. In particular, it is difficult to assess changes in the timing of peak flow given the RACMO data

resolution. Given the model mesh used here, where each lake consists of several mesh elements, a lake-focused analysis may be best served by a domain consisting of just one small lake basin to better resolve detailed processes such as changes in lake surface area and how this impacts the additional time lag imparted by the lake. For these considerations, we believe it's most appropriate to limit the current analysis to basin-scale features.

Channel density is spoken of throughout the entire manuscript yet there are no figures showing channel density evolution (only Figure 1)

References to "channel density" throughout the Introduction and elsewhere have been changed where the sentence is made more clear by instead referencing certain aspects of the channel network (e.g., Line 21, 35). We have added a sentence in the Model subsection to explain how local channel processes impact channel density (Line 59):

The density of supraglacial channels therefore changes as individual channel elements melt out if stream incision is insufficient to balance surface ablation.

**Minor Comments**

L2-6: Runon sentence

This sentence has been changed to explicitly enumerate the points being made:

We apply the Subaerial Drainage System (SaDS) model, a physically-based surface meltwater flow model, to a ~20 x 27 km$^2$ catchment on the southwestern Greenland Ice Sheet for four years of melt forcing (2011, 2012, 2015, and 2016) to (1) examine the relationship between surface melt rate and the rate, diurnal amplitude, and timing of surface inputs to moulins, (2) compare SaDS to contemporary models, and (3) present a framework for selecting appropriate supraglacial drainage models for different modelling objectives.

L4: change to "and the timing of surface meltwater inputs to moulins"

We appreciate the suggestion for clarity and precision, but since we have stated "surface meltwater inputs" twice already in the abstract, we believe the shorthand (surface inputs) is easily understood.

L13: Add citations to Smith et al., 2021, and Mejia et al., 2022. (Full citations at the end of this document).

We have added a citation to Smith et al. (2021) here and to Mejia et al. (2022) on Line 266 where we believe it is most relevant.

L16: Add citations to Yang et al., 2020. Added
L20: Define "efficient" here. We have added a definition for efficient "(i.e., faster)".

L21: Define what you mean here by "evolution of drainage density", the processes you describe typically control the evolution of a single channel (e.g., hydraulic capacity of that single channel), from the text it is not clear how these processes modify drainage density.

References to "drainage density" throughout the Introduction and elsewhere have been changed where the sentence is made more clear by instead referencing certain aspects of the channel network (e.g., Line 21, 35). We have added a sentence in the Model subsection to explain how local channel processes impact channel density (Line 59):

The density of supraglacial channels therefore changes as individual channel elements melt out if stream incision is insufficient to balance surface ablation.

L28-29: It is not clear what "supraglacial drainage characteristically acts to reduce the diurnal amplitude of moulin inputs" means, would the concentration of flow by supraglacial drainage systems not increase the amplitude of diurnal meltwater inputs to moulins?

What we intend to convey is that the diurnal amplitude of moulin inputs is typically much less than the diurnal amplitude of surface melt rates, and that the timing of peak moulin inputs lags the timing of peak surface melt rate. We have tried to make this clear by changing this sentence to (Line 28-29)

Water flow through the supraglacial drainage system characteristically acts to reduce the diurnal amplitude and delay the timing of moulin inputs relative to the diurnal cycle of surface melt [...]

L105: Do you mean diminished diurnal amplitude for smaller catchments? Over time? Be specific.

We mean diminished diurnal amplitude compared to surface melt rate (Line 109):

For all seven catchments, moulin inputs generally track surface melt rate (Fig. 2), diminished diurnal amplitude relative to the amplitude of the surface melt rate (Fig. 3) and a phase lag of ~2 to ~8 hours (Fig. 4e, f)

L105-111: consider combining paragraphs.

We believe it is easiest for the reader to keep these paragraphs separate since each paragraph describes a separate feature of the modelled drainage system.

L108-111:
L119: change & to "and" Changed here and throughout the manuscript.

L126: Figures 2—5 Corrected.
L128: $R^2$ is the coefficient of determination.
The reviewer is correct, and we have clarified this throughout. See also the response to the general comments.
L173-176: runon sentence

L172-177: Hard to understand paragraph
We have reorganized this paragraph to improve its clarity (Line 190-197):

Continuous seasonal trends in the amplitude and time lag of moulin inputs, as suggested by synthetic modelling (Yang et al., 2018; Hill and Dow, 2021), are not clear except in a few atypical cases. For example, in 2015, the diurnal amplitude of inputs to moulins M1--M5 steadily decreases with a statistically significant trend ($p<0.01$) from the onset of surface melting on 13 June until 2 July. Since this period (12 June to 2 July) is characterized by relatively steady surface melt rates (~1 to ~2 cm w.e. day$^{-1}$), this trend may be a result of a reduction in the extent of small supraglacial channels (Fig. B2). The end of the decreasing trend coincides with a rapid increase in melt rate from ~2 to ~4 cm w.e. day$^{-1}$

L295 (and elsewhere): I understand the use of the normalized or relative moulin input amplitude but this line is misleading, because the actual amplitude of moulin input variability is larger for your large catchments (it is only smaller/lower when you normalize by the overall larger discharge rates)

We have acknowledged this caveat here, and explained that the absolute diurnal amplitude is larger for the large catchments. The remaining references to diurnal amplitude are explicitly referenced as relative to the average moulin input rate (Line 320).

The four large catchments with supraglacial lakes within our domain have consistently lower relative diurnal amplitude in moulin inputs (however, a larger absolute diurnal amplitude given the larger magnitude of moulin inputs) [...]

Supplement
Figure S1: Please add a legend corresponding to the colors used in the plots as to not make the reader flip back and forth between the main text and the supplement. It is also not clear what you mean by bold colors vs. light colors, do you mean the black line here? Please make this more clear in either the legend or in the figure's caption. Further, it appears the colors used in the main text are different from those in the supplement.

We apologize for the confusion about the supplement. We had submitted a supplement as a companion to a previous version of the manuscript, however we no longer have a supplement associated with the current version. All additional figures are in the Appendix, and Table S1 is available as described in the code and data availability statement.

**References**
Andrews, L. C., Catania, G. A., Hoffman, M. J., Gulley, J. D., Lüthi, M. P., Ryser, C., Hawley, R. L., & Neumann, T. A. (2014). Direct observations of evolving subglacial drainage beneath the Greenland Ice Sheet. Nature, 514(7520), 80–83. https://doi.org/10.1038/nature13796

Smith, L. C., Andrews, L. C., Pitcher, L. H., Overstreet, B. T., Rennermalm, Å. K., Cooper, M. G., Cooley, S. W., Ryan, J. C., Miège, C., Kershner, C., & Simpson, C. E. (2021). Supraglacial River Forcing of Subglacial Water Storage and Diurnal Ice Sheet Motion. Geophysical Research Letters, 48(7). https://doi.org/10.1029/2020gl091418

Mejia, J. Z., Gulley, J., Trunz, C., Covington, M. D., Bartholomaus, T. C., Breithaupt, C. I., Xie, S., & Dixon, T. H. (2022). Moulin density controls the timing of peak pressurization within the Greenland Ice Sheet's subglacial drainage system. Geophysical Research Letters, 49, 1–13. https://doi.org/https://doi.org/10.1002/essoar.10511864.1

Yang, K., Sommers, A., Andrews, L. C., Smith, L. C., Lu, X., Fettweis, X., and Li, M. (2020)Intercomparison of surface meltwater routing models for the Greenland ice sheet and influence on subglacial effective pressures, The Cryosphere, 14, 3349–3365, https://doi.org/10.5194/tc-14-3349-2020.

Yang, K., & Smith, L. C. (2016). Internally drained catchments dominate supraglacial hydrology of the southwest Greenland Ice Sheet. Journal of Geophysical Research : Earth Surface, 121, 1891–1910. https://doi.org/doi:10.1002/ 2016JF003927

**Reviewer 2**

The authors have substantially revised and reorganized the manuscript following the review comments and I am satisfied with the new version of the manuscript. I find the new version well organized, complete and in sync with the title, abstract and introduction. The introduction and objective has been nicely rewritten. I am also pleased to see relevant additional tables and figures, and that the figures have been simplified and are now easier to read and interpret. The figure reorganization displaying the four years by data type makes sense. The model description in the appendix is a nice addition. The result and discussion section now thoroughly investigate changes in supraglacial discharge with drainage basin features for different melt years. The authors also interestingly compare their modeling results with a similar field site and provide relevant modeling recommendations.

We appreciate the reviewers comments and have responded individually below.

We would also like to highlight a change to the statistical analysis in Table 2. While investigating the first reviewer's comments to Line 126-136 and Table 2, we found a mistake in the calculation of $R^2$ and p values. This has been corrected, all other code has been double-checked, and Section 4.1 has been updated accordingly.

**Minor comments**
I have minor comments regarding the text:
Line 4: consider breaking the sentence after the parenthesis

This sentence has been changed to explicitly enumerate the points being made:

We apply the Subaerial Drainage System (SaDS) model, a physically-based surface meltwater flow model, to a ~20 x 27 km$^2$ catchment on the southwestern Greenland Ice Sheet for four years of melt forcing (2011, 2012, 2015, and 2016) to (1) examine the relationship between surface melt rate and the rate, diurnal amplitude, and timing of surface inputs to moulins, (2) compare SaDS to contemporary models, and (3) present a framework for selecting appropriate supraglacial drainage models for different modelling objectives.

L68-70: I was just wondering if you investigated how different smoothing of the ArcticDEM led to different routing and discharge results.

This is a good question. The 1.44 km moving average filter is the weakest smoothing for which we have achieved suitable numerical convergence. We agree that it is possible that modelled discharge is sensitive to the DEM smoothing we have applied. However, we believe the topography has not been overly smoothed, as evidenced by the persistence of supraglacial lakes and topographically controlled drainage pathways. Sub-grid scale roughness is likely to be an important control on moulin discharge, although this must be captured by the hydraulic

conductivity rather than the surface topography. We have explained the reason for the DEM smoothing following line 70:

We first smooth the ArcticDEM with a moving average filter with an edge length of 1.44 km, and then average the pixels that lie within each triangular element to define the centroid elevation. This smoothing is required to achieve numerical convergence within the SaDS model. It is possible that moulin inputs would change with higher resolution surface elevation data. However, it does not appear that the topography has not been overly smoothed here, as evidenced by the persistence of supraglacial lakes and topographically controlled drainage pathways.

L74: replace contraction "don't" by "do not". Same for L82. Done.

L103: is it a season average or a daily average that is used to calculate the "relative diurnal amplitude" ?

We use the seasonal average moulin input. This sentence has been clarified as follows (bolded text added; Line 105):

[...] the relative diurnal amplitude of inputs to moulins (measured as the peak-to-peak range in moulin inputs normalized by the **melt season-averaged** moulin input rate) (Fig. 3) [...]

L212: replace "it's" with "it is". Corrected.